# *Sphere2Vec*: Self-Supervised Location Representation Learning on Spherical Surfaces

## Abstract

Location encoding is valuable for a multitude of tasks where both the absolute positions and local contexts (image, text and other types of metadata) of spatial objects are needed for accurate predictions. However, most existing approaches do not leverage unlabeled data, which is crucial for use cases with limited labels. Furthermore, the availability of large-scale real-world GPS coordinate data demand representation and prediction at global scales. Existing location encoding models assume that the input coordinates are in Euclidean space, which can lead to modeling errors due to distortions introduced when mapping coordinates from other manifolds (e.g., spherical surfaces) to Euclidean space. We introduce *Sphere2Vec*, a location encoder, which can directly encode spherical coordinates while preserving spherical distances. *Sphere2Vec* is trained with a self-supervised learning framework which pre-trains deep location representations from unlabeled geo-tagged images with contrastive losses, and then fine-tunes to perform supervised geographic object classification tasks. *Sphere2Vec* achieves the performances of state-of-the-art results on various image classification tasks ranging from species, Point of Interest (POI) facade, to remote sensing. The self-supervised pretraining significantly improves the performance of *Sphere2Vec* especially when the labeled data is limited.

## 1 Introduction

Spatial information has become an important component to many machine learning tasks that address the environmental, economic, and societal needs for sustainable development. Just to name a few, these tasks include geo-aware species fine-grained recognition (Chu et al., 2019; Mac Aodha et al., 2019), Point of Interest (POI) facade image classification (Chu et al., 2019; Yan et al., 2018), remote sensing image classification (Christie et al., 2018; Ayush et al., 2020), poverty prediction (Jean et al., 2016; 2019), point cloud classification and semantic segmentation (Qi et al., 2017a;b), traffic forecasting (Li et al., 2018; Cai et al., 2020), and geographic question answering (Mai et al., 2019; 2020a). Developing a *general* model for vector space representations of any point in space, termed a *location encoder*, would pave the way for many future applications that improves our well-being. Such a location encoder can operate on a 2D space or on other manifolds; of particular interest is the spherical surface of the Earth.

Here we use geo-aware image classification shown in Figure 1 and 2 as an example. During training we assume a dataset $D = \{(I_i, \mathbf{x}_i, y_i)|i = 1, ..., N\}$, where $I_i$ is an image, $y_i \in 1, ...., C$ is the corresponding class label, $\mathbf{x}_i$ represents the location (longitude and latitude) and optionally the time the image was taken[1]. The image information is often processed by an state-of-the-art image classification model (e.g., InceptionV3) to predict the class labels as shown in Figure 1. Although the image itself is the most critical for image classification, sometimes it can be insufficient – the image of two distinct classes may look very similar both for the central object and its surrounding environment. For example, Figure 2a and 2f shows example images of two different fox species: Arctic fox (Vulpes lagopus) and bat-eared fox (Otocyon megalotis), for which the image model can be confused due to the high visual similarity of the two species. Fortunately, the geo-location information of an image may provide a clue to their real classes. Figure 2b, 2g show that these two species have distinct spatial distribution patterns, and the image locations provide complementary information to the images. This

---

[1]In this study we focus on the location information and leave the time aspect of modeling to future.

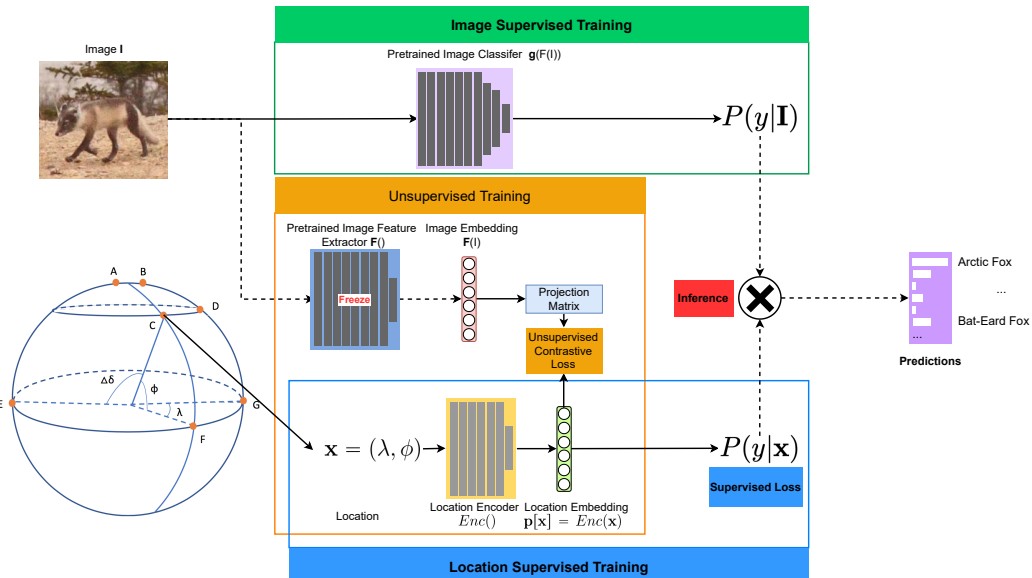

Figure 1: Applying *Sphere2Vec* to geo-aware image classification. The supervised image classification module (green) is a InceptionV3 network similar to that of Mac Aodha et al. (2019); the supervised location classification module (blue) can be *Sphere2Vec* or any other inductive location encoders (Chu et al., 2019; Mac Aodha et al., 2019; Mai et al., 2020b); the self-supervised training stage (orange) pre-trains the location encoder based on unlabeled image data. The dotted lines indicates that there is no back-propergation through these lines.

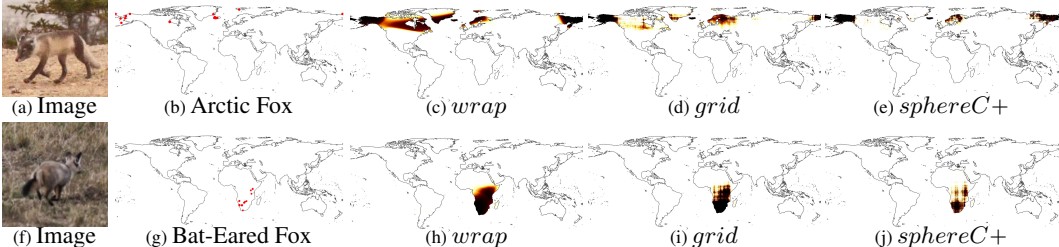

Figure 2: Applying location encoders to differentiate two visually similar species: Arctic fox and bat-eared fox. (a) and (f) are their example images which look very similar. (b) and (g) show their distinct patterns in image locations. (c)-(e): The predicted distributions of Vulpes lagopus from different location based models (without images as input). (h)-(j): The predicted distributions of Otocyon megalotis. While $wrap*$ (Mac Aodha et al., 2019) produces a over-generalized species distribution, $sphereC+$ (our model) produces a more compact and fine-grain distribution on the polar region and in data sparse areas such as Africa (See Figure 2g-2j). $grid$ (Mai et al., 2020b) is between the two. For more examples, please see Figure 10.

insight motivates the development of hybrid classification models, which take both the images and their locations as input (Mac Aodha et al., 2019; Mai et al., 2020b).

In spite of their effectiveness, hybrid models have an issue when representing points in non-Euclidean spaces. The demand on representation and prediction learning at a global scale grows dramatically due to emerging global scale issues, such as the transition path of the latest pandemic (Chinazzi et al., 2020), long lasting issue for malaria (Caminade et al., 2014), under threaten global biodiversity(Di Marco et al., 2019; Ceballos et al., 2020), and numerous ecosystem and social system responses for climate change (Hansen & Cramer, 2015). However, when applying the state-of-the-art 2D space location encoders (Gao et al., 2019; Chu et al., 2019; Mac Aodha et al., 2019; Mai et al., 2020b; Zhong et al., 2020) to large-scale real-world GPS coordinate datasets such as species images taken all over the world, a **map projection distortion problem** (Williamson & Browning, 1973) emerges. These 2D location encoders are designed for preserving distance in 2D (or 3D) Euclidean space, while GPS coordinates are in fact on a spherical manifold, e.g., point $\mathbf{x} = (\lambda, \phi)$ on a sphere with longitude $\lambda \in [-\pi, \pi)$ and latitude $\phi \in [-\pi/2, -\pi/2]$ (See Figure 1). Directly applying these 2D location encoders on spherical coordinates will yield a large distortion in the polar regions. Note that *map projection distortion is unavoidable when projecting spherical coordinates into 2D space.* This emphasizes the importance of *calculating on a round planet* (Chrisman, 2017). See Appendix 9.1 for more discussion.

From a modeling perspective another key challenge is how to design the self-supervised training objective that effectively transfers the attribute information (e.g., image features) at different locations into a location encoder. This is very important in practice since collecting and labeling geo-tagged images is particularly costly as annotations often require domain expertise. However, most existing location encoding approaches (Chu et al., 2019; Mac Aodha et al., 2019; Mai et al., 2020b) are developed and trained in a supervised learning framework while massive unlabeled geographic data cannot be used. Ayush et al. (2020) proposed a geographic self-supervised learning framework, but they mainly focus on training image encoders while the usage of geo-location information is limited to data augmentation in the pre-training stage. In contrast, in this work, we focus on training location encoders in a self-supervised manner so that geo-location information is explicitly use as part of the model input.

In this work, we propose *Sphere2Vec* as shown in Figure 1, which directly encodes point coordinates on a spherical surface while avoiding the map projection distortion. The multi-scale encoding method utilizes Double Fourier Sphere basis ($O(S^2)$ terms) or a subset ($O(S)$ terms) of it while still being able to correctly measure the spherical distance. Furthermore, we develop a self-supervised learning framework which pre-trains deep location representations from unlabeled geo-tagged images by predicting image features or image identities base on their geo locations. We compare several self-supervised learning objectives including Mean Square Error loss (MSE), contrastive binary classification loss (BI), and contrastive multi-classification loss (MC). The pre-trained *Sphere2Vec* model can be fine-tuned (with an additional output layer) to perform supervised geographic objects classification tasks. We demonstrate the effectiveness of *Sphere2Vec* on geo-aware image classification tasks including fine-grain species recognition (Chu et al., 2019; Mac Aodha et al., 2019; Mai et al., 2020b), POI fascade image classification (Tang et al., 2015; Mac Aodha et al., 2019), and remote sensing image recognition (Christie et al., 2018; Ayush et al., 2020).

**In summary, the contributions of our work are:**

1. We develop a unified view of distant preserving location encoding methods on spheres based on Double Fourier Sphere (DFS) (Merilees, 1973; Orszag, 1974).
2. We propose an effective pre-training strategy for location encoders from unlabeled geo-tagged images by predicting image features and image identities based on their geo locations.
3. *Sphere2Vec* improves the performances of state-of-the-art results on four species recognition datasets (BirdSnap, NABirds, iNat2017, iNat2018), one POI fascade image classification dataset (YFCC), and one remote sensing image classification dataset (fMoW). It also demonstrate significant advantage in few shot learning settings.

## 2 PROBLEM FORMULATION

Following previous work (Mai et al., 2020b), *Euclidean location encoding* can be formulated as follows. Given a set of points (e.g., POI) $\mathcal{P} = \{p_i\}$ in $L$-D space ($L = 2, 3$), we define a function $e_{\mathcal{P}, \theta}(\mathbf{x}) : \mathbb{R}^L \to \mathbb{R}^d$ ($L \ll d$), which is parameterized by $\theta$ and maps any coordinate $\mathbf{x}$ in space to a vector representation of $d$ dimension. Each point (e.g., a restaurant) $p_i = (\mathbf{x}_i, \mathbf{v}_i)$ is associated with a location $\mathbf{x}_i$ and attributes $\mathbf{v}_i$ (i.e., POI features such as types, names, capacity, images, etc.). The function $e_{\mathcal{P}, \theta}(\mathbf{x})$ encodes the information about Point $\mathbf{x}$ which is useful for certain tasks such as predicting a label $\mathbf{y}$. Attributes and coordinates of points can be seen as analogies to words and word positions in commonly used word embedding models.

Similarly, we can formulate *spherical location encoding* as follows. Given a set of points $\mathcal{P} = \{p_i\}$ on the surface of a sphere $\mathbb{S}^2$, e.g., species occurrences all over the world, where $p_i = (\mathbf{x}_i, \mathbf{v}_i)$ and $\mathbf{x}_i = (\lambda_i, \phi_i) \in \mathbb{S}^2$ indicates a point with longitude $\lambda_i \in [-\pi, \pi)$ and latitude $\phi_i \in [-\pi/2, \pi/2]$. Define a function $e_{\mathcal{P}, \theta}(\mathbf{x}) : \mathbb{S}^2 \to \mathbb{R}^d$, which is parameterized by $\theta$ and maps any coordinate $\mathbf{x}$ in a spherical surface $\mathbb{S}^2$ to a vector representation of $d$ dimension. See Appendix 9.1 for more discussions about the map projection distortion problem.

## 3 METHOD

The *geo-aware image classification task* (Chu et al., 2019; Mac Aodha et al., 2019) can be formulated as follows. Given an image $\mathbf{I}$ taken from location/point $\mathbf{x}$, we estimate which category $y$ it belongs to by $P(y|\mathbf{I}, \mathbf{x})$. If we assume $\mathbf{I}$ and $\mathbf{x}$ are conditionally independent given $y$, then based on Bayes'

theorem, we have $P(y|\mathbf{I}, \mathbf{x}) \propto P(y|\mathbf{x})P(y|\mathbf{I})$ (See Mac Aodha et al. (2019)). $P(y|\mathbf{I})$ can be given by any state-of-the-art image classifier such as Inception V3 (Szegedy et al., 2016) for the species recognition task or MoCo-V2+TP (Ayush et al., 2020) for the RS image classification task. Figure 1 illustrates the whole training and inference process by using species recognition as the example task. $P(y|\mathbf{I})$ and $P(y|\mathbf{x})$ are given by the image encoder (the green box) and location encoder (the blue box) respectively and are multiplied together for final prediction. In this work, we focus on estimating the geographic prior distribution of class $y$ over the spherical surface $P(y|\mathbf{x}) \propto \sigma(e(\mathbf{x})\mathbf{T}_{:,y})$ where $\sigma()$ is a sigmoid activation function. $\mathbf{T} \in \mathbb{R}^{d \times c}$ is a class embedding matrix where the $y_{th}$ column $\mathbf{T}_{:,y} \in \mathbb{R}^{d}$ indicates the class embedding for class $y$.

In the following we introduce the two main contributions of our work – location encoding on spherical surfaces (the blue box in Figure 1), and a self-supervised training framework for location representation model (the orange box in Figure 1). In Section 3.1, we will discuss the design of spherical distance-kept location encoder $e(\mathbf{x})$, *Sphere2Vec*. We developed a unified view of distance-reserving encoding on spheres based on Double Fourier Sphere (DFS) (Merilees, 1973; Orszag, 1974). In Section 3.2 we discuss training strategies for *Sphere2Vec* including self-supervised pre-training given unlabeled images, and supervised fine tuning. We use $e_{\mathcal{P},\theta}(\mathbf{x})$ to indicate the spherical location encoding neural network with parameters $\theta$, and will refer it by $e(\mathbf{x})$ for simplicity.

### 3.1 *Sphere2Vec* ENCODER

We want to find a function $G^{(sphere)}(\mathbf{x})$ which does a one-to-one mapping from each point $\mathbf{x}_i = (\lambda_i, \phi_i) \in \mathbb{S}^2$ to a multi-scale representation with $S$ as the total number of scales such that it satisfies the following requirement:

$$\langle G^{(sphere)}(\mathbf{x}_1), G^{(sphere)}(\mathbf{x}_2)\rangle = f(\Delta D), \forall \mathbf{x}_1, \mathbf{x}_2 \in \mathbb{S}^2, \tag{1}$$

where $\Delta D \in [0, \pi R]$ is the spherical surface distance between $\mathbf{x}_1, \mathbf{x}_2$, $R$ is the radius of this sphere, and $f(x)$ is a strictly monotonically decreasing function for $x \in [0, \pi R]$. In other words, we expect to find a function $G^{(sphere)}(\mathbf{x})$ such that the resulting multi-scale representation of $\mathbf{x}$ preserves the spherical surface distance.

Similar to the case for Euclidean space (Mai et al., 2020b), any point $\mathbf{x} = (\lambda, \phi) \in \mathbb{S}^2$ with longitude $\lambda \in [-\pi, \pi)$ and latitude $\phi \in [-\frac{\pi}{2}, \frac{\pi}{2}]$, can be explicitly encoded in a multi-scale fashion in the form of $e^{(x)}(\mathbf{x}) = \mathbf{NN}_{ffn}(G^{(*)}(\mathbf{x}))$ where $\mathbf{NN}_{ffn}()$ is a learnable multi-layer perceptron with $h$ hidden layers and $k$ neurons per layer. $G^{(*)}(\mathbf{x})$ is a concatenation of multi-scale spherical spatial features of $S$ levels. $(*)$ indicates different types of spherical location encoders. In the following, we call $e^{(x)}(\mathbf{x})$ *location encoder* and its component $G^{(*)}(\mathbf{x})$ *position encoder*. Let $r_{min}, r_{max}$ be the minimum and maximum scaling factor, and $g = \frac{r_{max}}{r_{min}}$.[2]

**sphereDFS** Double Fourier Sphere (DFS) (Merilees, 1973; Orszag, 1974) is a simple yet successful pseudospectral method, which has been applied to efficient analysis of large scale phenomenons such as weather (Sun et al., 2014) and blackholes (Bartnik & Norton, 2000). Our first intuition is to use the base functions of DFS to help decompose $\mathbf{x} = (\lambda, \phi)$ into a high dimensional vector:

$$G^{(sphereDFS)}(\mathbf{x}) = \bigcup_{n=0}^{S-1} [\sin \phi_n, \cos \phi_n] \cup \bigcup_{m=0}^{S-1} [\sin \lambda_m, \cos \lambda_m] \cup$$
$$\bigcup_{n=0}^{S-1} \bigcup_{m=0}^{S-1} [\cos \phi_n \cos \lambda_m, \cos \phi_n \sin \lambda_m, \sin \phi_n \cos \lambda_m, \sin \phi_n \sin \lambda_m]. \tag{2}$$

where $\forall s = 0, 1, ..., S-1$, we define $\lambda_s = \frac{\lambda}{f_s}$, $\phi_s = \frac{\phi}{f_s}$, $f_s = r_{min} \cdot g^{s/(S-1)}$. $\cup$ means vector concatenation and $\bigcup_{s=0}^{S-1}$ indicates vector concatenation through different scales. It basically lets all the scales of $\phi$ terms interact with all the scales of $\lambda$ terms in the encoder. This would introduce a position encoder whose output has $O(S^2)$ dimensions and increase the memory burden in training and hurts generalization (Figure 3a).

---

[2]In practice we fix $r_{max} = 1$ meaning no scaling of $\lambda, \phi$.

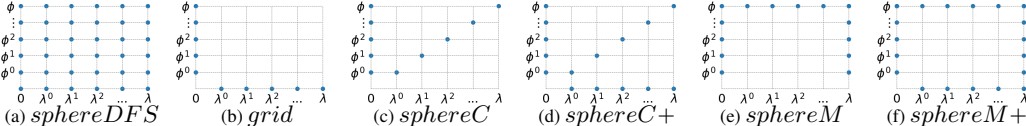

Figure 3: Spectral representations of different encoders. Each point $(\lambda^m, \phi^n)$ represent an interaction terms of trigonometric functions in the encoder. Points on the axis correspond to single terms with no interactions.

An encoder might achieve better result by only using a subset of these terms. In comparison, the state of art encoder such as $grid$ and $wrap$ (Mac Aodha et al., 2019) define: [3]

$$G^{(grid)}(\mathbf{x}) = \bigcup_{s=0}^{S-1} [\sin \phi_s, \cos \phi_s, \sin \lambda_s, \cos \lambda_s]. \tag{3}$$

We can see that $grid$ employs a subset of terms from $sphereDFS$ (Figure 3b) and performs poorly at a global scale due to its inability to preserve spherical distances (See Appendix 9.3 for proof).

We explore different subsets of DFS terms while achieving two goals: 1) efficient representation with $O(S)$ dimensions 2) preserving distance measures on spheres. See Appendix 9.2 for the detail definition of $sphereC$, $sphereC+$, $sphereM$, $sphereM+$, and Figure 3 for a unified view. We hypothesize that encoding these terms in the multi-scale representation would make the training of the encoder easier and the order of output dimension is still $O(S)$.

In location encoding, the uniqueness of encoding (no two points on sphere having the same position encoding) is very important, $G^{(*)}(\mathbf{x})$ in the five proposed methods are one-to-one mapping.

**Theorem 1.** $\forall * \in \{sphereC, sphereC+, sphereM, sphereM+, sphereDFS\}$, $G^{(*)}(\mathbf{x})$ *is an injective function. See the proof in Appendix 9.5.*

### 3.2 TRAINING *Sphere2Vec*

**Self-Supervized Pretraining (the yellow box in Figure 1).** We consider two contrastive objectives. The first is a binary classification loss, or so called *noise contrastive estimation* (Gutmann & Hyvärinen, 2010), which avoids calculation of the partition function and has been successfully used in word embeddings (Mikolov et al., 2013) and language modeling (Mnih & Teh, 2012):

$$\mathcal{L}_{\mathrm{BI}}(\mathcal{P}, \mathcal{N}) = -\mathbb{E}_{(\mathbf{a}, \mathbf{b}) \sim \mathcal{P}}[\log \sigma(s(\mathbf{a}, \mathbf{b}))] - \mathbb{E}_{(\mathbf{a}, \mathbf{b}^-) \sim \mathcal{N}}[\log(1 - \sigma(s(\mathbf{a}, \mathbf{b}^-)))] \tag{4}$$

where BI stands for "binary", $\mathcal{P}$ is a set of positive pairs (denoted as $(\mathbf{a}, \mathbf{b})$), $\mathcal{N}$ is a set of negative pairs (denoted as $(\mathbf{a}, \mathbf{b}^-)$), $s(\cdot, \cdot)$ is a similarity function that assigns higher values when the inputs are more similar (such as $cosine()$), and $\sigma(v) = e^v/(1 + e^v)$ is the sigmoid function.

The second objective function is the multi-class classification loss with temperature, which has been successfully used in unsupervised learning for images (He et al., 2020) and text (Gao et al., 2021):

$$\mathcal{L}_{\mathrm{MC}}(\mathcal{P}, \mathcal{N}, \tau) = \mathbb{E}_{(\mathbf{a}, \mathbf{b}) \sim \mathcal{P}} \left[ \frac{\exp(s(\mathbf{a}, \mathbf{b})/\tau)}{\exp(s(\mathbf{a}, \mathbf{b})/\tau) + \sum_{\mathbf{b}^- \in \mathcal{N}(\mathbf{b}|\mathbf{a})} \exp(s(\mathbf{a}, \mathbf{b}^-)/\tau)} \right] \tag{5}$$

where MC stands for "multi-class", $\mathcal{N}(\mathbf{b}|\mathbf{a})$ obtains a negative pair with first entry being $\mathbf{a}$; $\mathcal{P}$ and $s(\cdot, \cdot)$ are defined as earlier. The temperature scaling parameter $\tau$ determines how soft the softmax is (Hinton et al., 2015). In practice it help with the trade off between top ranked classes (precision) versus reset of the classes (recall).

In order to learn useful representations, we need to choose appropriate distributions for positive pairs $\mathcal{P}$ and negative pairs $\mathcal{N}$. During the location encoder pretraining stage we use a geo-tagged but unlabeled image set $\mathbb{X}^{unlabeled}$, each image is represented by a embedding generated from a pretrained image extract extractor $\mathbb{F}$[4] such as InceptionV3 or MoCo-V2+TP (Ayush et al., 2020).

Given an unlabeled image $(\mathbf{x}, \mathbf{I}) \in \mathbb{X}^{unlabeled}$, we use the following positive and negative instances:

---

[3] As a multi-scale encoder, $grid$ degenerates to $wrap$ when the number of scales $S = 1$.

[4] $\mathbb{F}$ should not see the current image labels during pretraining stage.

- **Geo-tagged positive** $\mathcal{P}^x = \{(e(\mathbf{x}), \mathbf{W}\mathbb{F}(\mathbf{I}))\}$ where $\mathbf{W}$ is a linear projection layer.
- **In-batch negatives** $\mathcal{N}^b = \{(e(\mathbf{x}), \mathbf{W}\mathbb{F}(\mathbf{I}^-)|(\mathbf{x}^-, \mathbf{I}^-) \in \mathcal{B}(\mathbf{x}, \mathbf{I})\backslash\{(\mathbf{x}, \mathbf{I})\}\}$ where $\mathcal{B}(\mathbf{x}, \mathbf{I})$ represents the batch of examples for which $(\mathbf{x}, \mathbf{I})$ is part of during training.
- **Sampled negative locations** $\mathcal{N}^s = \{(e(\mathbf{x}^-), \mathbf{W}\mathbb{F}(\mathbf{I}))\}$ of size $N$, where $\mathbf{x}^-$ is one of the $N$ evenly sampled locations from the surface of the sphere for each example $\mathbf{x}$.
- **Dropout positive** $\mathcal{P}^d = \{(e(\mathbf{x}), e'(\mathbf{x}))\}$, where we pass the same input $\mathbf{x}$ to the encoder $e()$ twice and obtain two embeddings as "positive pairs" by applying independently sampled dropout masks. This is a data augmentation strategy (so called SimCSE), which has been very successful in generating sentence embeddings (Gao et al., 2021). We use $e'(\mathbf{x})$ to denote the embedding from the second mask.
- **Dropout negative** $\mathcal{N}^d = \{(e(\mathbf{x}), e'(\mathbf{x}^-))|(\mathbf{x}^-, \mathbf{I}^-) \in \mathcal{B}(\mathbf{x}, \mathbf{I})\backslash\{(\mathbf{x}, \mathbf{I})\}$, where $e'(\cdot)$ has the same meaning as that defined in $\mathcal{P}^d$ and $\mathcal{B}(\mathbf{x}, \mathbf{I})$ indicates the same as $\mathcal{N}^b$.

We define two versions of contrastive losses, which both have three components: in-batch, negative location, and SimCSE. *The self-supervized binary classification loss $\mathcal{L}_{\mathrm{BI}}$ is defined as*

$$
\begin{aligned}
\mathcal{L}_{\mathrm{BI}}(\mathbb{X}) &= \mathcal{L}_{\mathrm{BI}}^{batch}(\mathbb{X}) + \beta_1 \mathcal{L}_{\mathrm{BI}}^{negloc}(\mathbb{X}) + \beta_2 \mathcal{L}_{\mathrm{BI}}^{simcse}(\mathbb{X}) \\
&= \mathcal{L}_{\mathrm{BI}}(\mathcal{P}^x, \mathcal{N}^b) + \beta_1 \mathcal{L}_{\mathrm{BI}}(\varnothing, \mathcal{N}^s) + \beta_2 \mathcal{L}_{\mathrm{BI}}(\mathcal{P}^d, \mathcal{N}^d)
\end{aligned}
\tag{6}
$$

where $\beta_1$ and $\beta_2$ control the contribution of the last two loss components.

*The self-supervized multi-class classification loss $\mathcal{L}_{\mathrm{MC}}$ is defined as*

$$
\begin{aligned}
\mathcal{L}_{\mathrm{MC}}(\mathbb{X}) &= \mathcal{L}_{\mathrm{MC}}^{batch}(\mathbb{X}) + \alpha_1 \mathcal{L}_{\mathrm{MC}}^{negloc}(\mathbb{X}) + \alpha_2 \mathcal{L}_{\mathrm{MC}}^{simcse}(\mathbb{X}) \\
&= \mathcal{L}_{\mathrm{MC}}(\mathcal{P}^x, \mathcal{N}^b, \tau_0) + \alpha_1 \mathcal{L}_{\mathrm{MC}}(\mathcal{P}^x, \mathcal{N}^s, \tau_1) + \alpha_2 \mathcal{L}_{\mathrm{MC}}(\mathcal{P}^d, \mathcal{N}^d, \tau_2)
\end{aligned}
\tag{7}
$$

where $\alpha_1$ and $\alpha_2$ are hyper-parameters.

**Supervised Fine-Turning (the blue box in Figure 1).** We directly use image labels in the training objective. Following Mac Aodha et al. (2019), we used a *presence-absence loss* function which converts the multi-class labels into binary multi-labels. Given a set of training samples $\mathbb{X}^{labeled} = \{(\mathbf{x}, y)\}$, the loss function $\mathcal{L}^{supervized}(\mathbb{X})$ is defined as:

$$
\mathcal{L}^{supervized}(\mathbb{X}) = \beta \mathcal{L}_{\mathrm{BI}}(\mathcal{P}^l, \varnothing) + \mathcal{L}_{\mathrm{BI}}(\varnothing, \mathcal{N}^l) + \mathcal{L}_{\mathrm{BI}}(\varnothing, \mathcal{N}^{ls})
\tag{8}
$$

Here, $\beta$ is a hyperparameter to enlarge the weight of positive samples, and the following positive and negative samples are used:

- **Labeled positives** $\mathcal{P}^l = \{(e(\mathbf{x}), \mathbf{T}_{:,y})|(\mathbf{x}, y) \in \mathbb{X}\}$.
- **Labeled negatives** $\mathcal{N}^l = \{(e(\mathbf{x}), \mathbf{T}_{:,i})|(\mathbf{x}, y) \in \mathbb{X}, i \in \{1..c\}\backslash\{y\}\}$.
- **Sampled negative locations** $\mathcal{N}^{ls} = \{(e(\mathbf{x}^-), \mathbf{T}_{:,i})|(\mathbf{x}, y) \in \mathbb{X}, i \in \{1..c\}\}$, where $\mathbf{x}^-$ is one of the $N$ evenly sampled locations from the surface of the sphere for each example $\mathbf{x}$.

## 4 RELATED WORK

**Multi-Scale Position/Location Encoding on Spheres** Recent works in various applications (Zhong et al., 2020; Mai et al., 2020b; Mildenhall et al., 2020; Tancik et al., 2020) found that a simple sinusoidal mapping of input coordinates ("positional encoding") allows multilayer perceptrons (MLP) to represent higher frequency content. More specifically $f_\theta$ is decomposed into $f_\theta = \mathbf{NN}_\theta(PE(\mathbf{x}))$, where $PE$ is a deterministic function mapping $\mathbf{x}$ from $R^d$ into higher dimensional space and $\mathbf{NN}_\theta$ is simply a MLP. This strategy is compelling since MLPs can be orders of magnitude more compact than grid-sampled representations in order to achieve similar order of details (Tancik et al., 2020). They show that passing input points through a simple Fourier feature mapping enables a MLP to learn high-frequency functions in low-dimensional problem domains. This development is consistent with Rahaman et al. (2019a) who show that deep networks are biased towards learning lower frequency functions, and mapping the inputs to a higher dimensional space using high frequency functions enables better fitting of data that contains high frequency details. In this work we extend the success of this line of work to none Euclidean spaces such as spherical surface. We leverage truncated Fourier transformation on spheres to achieve computation efficiency while avoiding the error caused by projection distortion.

**Unsupervised Representation Learning** Unsupervised text encoding models such as transformer (Vaswani et al., 2017; Devlin et al., 2019) has been effectively utilized in many Natural Language Processing (NLP) tasks. At its core, a trained model encodes words into vector space representations based on their positions and context in the text. Following the success in NLP, there has been significen recent progress in unsupervised image pretraining (He et al., 2020; Caron et al., 2020; Ayush et al., 2020). Interestingly almost all of them are based on certain form of contrastive learning (Hadsell et al., 2006), which helps to construct unsupervised classification objectives from continuous inputs such as images. He et al. (2020) proposes Momentum Contrast (MoCo) for unsupervised visual representation learning. To increase the number of negative examples in contrastive training, they uses a queue of multiple mini-batches. Similar strategy has been adopted in NLP (Gao et al., 2021). To improve the encoding inconsistency between mini batches, they make the target image encoder parameterizes a moving average of the query image encoder. In this work we are focusing on the pretraining of location encoder with a frozen image encoder. Our approach is very memory efficient (easily scaling up to 1024 batch size) and therefore avoid the need of multi-batch training. Ayush et al. (2020) combines a geo-location classification loss and a contrastive image loss similar to MoCo during pre-training. For geo-location classification, a deep image network is trained to predict a coarse geo-location of where in the world the image might come from. Since the geo-location is not part of the classifier's input, it is hard for the model to directly leverage it in the prediction process. Zhai et al. (2018) learn location representation from unlabeled image-location pairs for image localization. They apply cross entropy loss to discretized location (or time), which cannot leverage the continuity of the approximated function. One main difference and also one of our main contributions is our contrastive self-supervised loss, which avoids the need of discretization.

## 5 EXPERIMENT

### 5.1 THE EFFECTIVENESS OF SPHERICAL LOCATION ENCODING

To test the effectiveness of *Sphere2Vec* we first conduct geo-aware image classification experiments in a full supervised setting (without the self-supervised pretraining step) on seven large-scale real-world datasets including 5 (animal) species datasets, one POI dataset (YFCC), and one remote sensing dataset (fMoW). Please refer to Mac Aodha et al. (2019) and Christie et al. (2018) for the dataset descriptions. Table 1 compares the result of five variants of *Sphere2Vec* models against nine baseline models (see their detailed descriptions in Appendix 9.6. We can see that the *Sphere2Vec* models outperform baselines on all seven datasets, and the variants with linear number of DFS terms works as well as or even better than $sphereDFS$. Figure 2c-2e and 2h-2j show the predicted species distributions from different models for two fox species. We can see that $wrap$ and $grid$ have overgeneralization issue in data sparse area (e.g., Africa) and the North Pole area. The distributions produced by $sphereC+$ are more compact and generalizable.

To help understand the impact of *Sphere2Vec*, we conduct detailed analysis on iNat2017 and iNat2018. Figure 4a shows the image locations in iNat2017 validation dataset. We split this dataset into different latitude bands and compare the $\Delta MRR$ between each model to $grid$. Figure 4b and 4c show the number of samples and $\Delta MRR$ in each band while Figure 4f shows that the contrast between these two variables for different models. We can see that *Sphere2Vec* models have larger $\Delta MRR$ at the North Pole region (Figure 4c). Moreover, *Sphere2Vec* has advantages on bands with less data samples, e.g. $\phi \in [-30°, -20°]$. We also compare $sphereC+$ and $grid$ at a quantized spatial resolution - in latitude-longitude cells(Figure 4d and 4e). $sphereC+$ has more advantages in North Pole and data sparse cells. For more details please refer to Appendix 9.12, 9.13, and 9.14.

### 5.2 THE EFFECTIVENESS OF SELF-SUPERVISED PRETRAINING

Next, we investigate the impact of self-supervised contrastive pretraining (see Section 3.2). For species recognition and POI image classification task, we use the pretrained Inception-V3 model provided by PyTorch as image feature extractor $\mathbb{F}$ in the yellow box of Figure 1. As for fMoW dataset, we use the unsupervised pretrained MoCo-V2+TP (Ayush et al., 2020) as $\mathbb{F}$. Both models are supervised fine-tuned on the respective dataset to act as $g(\mathbb{F}(\cdot))$ in the green box of Figure 1. We consider three self-supervised training objectives - mean square error loss MSE between the location embedding $e(\mathbf{x})$ and the image feature $\mathbb{F}(\mathbf{I})$, noise contrastive binary classification loss BI (see Equation 6), and contrastive multi-class classification loss MC (See Equation 7). In order to

Table 1: Geo-aware image classification under fully supervised settings over three tasks: species recognition, POI image classification (YFCC), and remote sensing (RS) image classification (fMOW (Christie et al., 2018)). *tile* indicates the results reported by Mac Aodha et al. (2019). *wrap∗* indicates the original results reported by Mac Aodha et al. (2019) while *wrap* is the best results we obtained by rerunning their code. See Appendix 9.6 for details about these baselines. Since the test sets for iNat2017 and iNat2018 are not open-sourced, we report results on validation sets. The best performance of the baseline models and *Sphere2Vec* are highlighted as bold. All compared models use location only while ignoring time. The original result reported by Ayush et al. (2020) for No Prior on fMOW is 69.05. We obtain 69.84 by retraining their implementation. Here we report Top1 accuracy. See Appendix 9.9 for more results. See Appendix 9.8.1 for hyperparameter tuning details.

| Task | Species Recognition | | | | | | POI | RS |
|---|---|---|---|---|---|---|---|---|
| Dataset | BirdSnap | BirdSnap† | NABirds† | iNat2017 | iNat2018 | Avg | YFCC | fMOW |
| P(y\|x) - Prior Type | Test | Test | Test | Val | Val | - | Test | Val |
| No Prior (i.e. image model) | 70.07 | 70.07 | 76.08 | 63.27 | 60.2 | 67.94 | 50.15 | 69.84 |
| *tile* (Tang et al., 2015) | 70.16 | 72.33 | 77.34 | 66.15 | 65.61 | 70.32 | 50.43 | - |
| *xyz* | 71.85 | 78.97 | 81.2 | 69.39 | 71.75 | 74.63 | 50.75 | 70.18 |
| *wrap∗* (Mac Aodha et al., 2019) | 71.66 | 78.5 | 81.15 | 69.34 | 72.41 | 74.64 | 50.70 | - |
| *wrap* | 71.87 | 79.06 | **81.62** | 69.22 | 72.92 | 74.94 | 50.90 | 70.29 |
| *wrap + ffn* | **71.99** | 79.21 | 81.36 | 69.4 | 71.95 | 74.78 | 50.76 | 70.28 |
| *rbf* (Mai et al., 2020b) | 71.78 | 79.4 | 81.32 | 68.52 | 71.35 | 74.47 | 51.09 | 70.65 |
| *rff* (Rahimi et al., 2007) | 71.92 | 79.16 | 81.3 | 69.36 | 71.8 | 74.71 | 50.67 | 70.27 |
| *grid* (Mai et al., 2020b) | 71.7 | **79.44** | 81.24 | 69.46 | 72.98 | 74.96 | **51.18** | 70.81 |
| *theory* (Mai et al., 2020b) | 71.88 | 79.24 | 81.59 | **69.47** | **73.01** | **75.04** | 51.16 | 70.82 |
| *sphereC* | 72.06 | 79.8 | 81.86 | 69.62 | 73.29 | 75.33 | 51.34 | 71.00 |
| *sphereC+* | **72.41** | 80.02 | 81.81 | **69.66** | 73.31 | 75.44 | 51.28 | 71.03 |
| *sphereM* | 72.02 | 79.75 | 81.69 | **69.69** | 73.25 | 75.28 | **51.35** | 70.98 |
| *sphereM+* | 72.24 | **80.34** | **81.9** | 69.67 | **73.72** | **75.57** | 51.24 | 71.10 |
| *sphereDFS* | 71.75 | 79.18 | 81.39 | 69.65 | 73.24 | 75.04 | 51.15 | **71.46** |

(a) Validation Locations  (b) Samples per $\phi$ band  (c) $\Delta MRR$ per $\phi$ band

(d) $\Delta MRR$ per cell  (e) $\Delta MRR$ per cell  (f) $\Delta MRR$ per $\phi$ band

Figure 4: iNat2017 dataset and model performance comparison: (a) Sample locations for validation set where the dashed and solid lines indicates latitudes; (b) The number of training and validation samples in different latitude intervals. (c) $MRR$ difference between a model and baseline *grid* on the validation dataset. (d) $\Delta MRR = MRR(sphereC+) - MRR(grid)$ for each latitude-longitude cell. Red and blue color indicates positive and negative $\Delta MRR$ while darker color means high absolute value. The number on each cell indicates the number of validation data points while "1K+" means there are more than 1K points in a cell. (e) The number of validation samples v.s. $\Delta MRR = MRR(sphereC+) - MRR(grid)$ per latitude-longitude cell. The orange dots represent moving averages. (b) The number of validation samples v.s. $\Delta MRR$ per latitude band.

simulate the situations with limited labeled data we first perform self-supervised pretraining for $e(\mathbf{x})$ with one of the three objectives using 100% of the unlabeled dataset $\mathbb{X}^{unlabeled}$ (e.g., iNatlist 2018). Then we perform stratified sampling to obtain $\Gamma$ portion of the labeled data, denoted as $\mathbb{X}_\Gamma$, while keeping its label distribution the same as the full training set. Finally we fine-tune the pretrained $e(\mathbf{x})$ on $\mathbb{X}_\Gamma$ with the supervised learning objective (Equation 8).

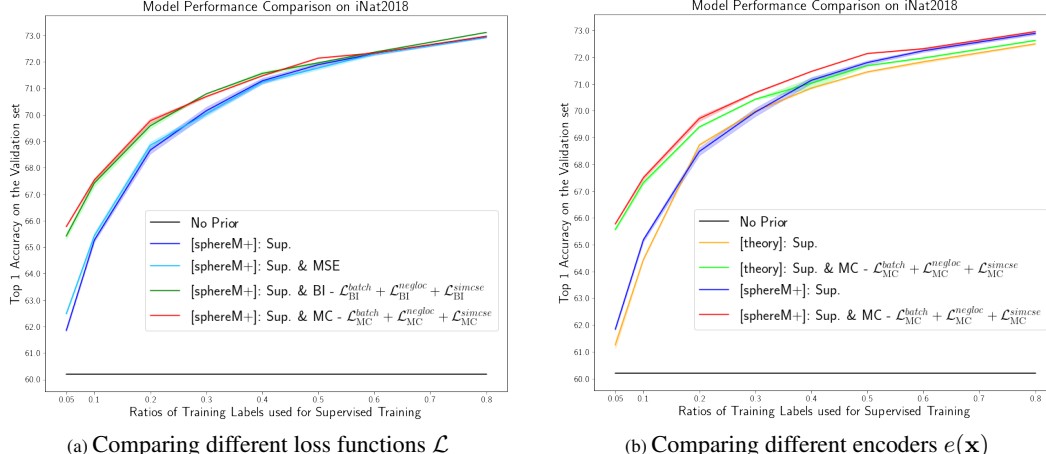

(a) Comparing different loss functions $\mathcal{L}$      (b) Comparing different encoders $e(\mathbf{x})$

Figure 5: Comparing different self-supervised learning objectives $\mathcal{L}$ and different encoders $e(\mathbf{x})$ in a label limited setting on iNat2018. $e(\mathbf{x})$ is first trained on $\mathbb{X}^{unlabeled}$ with different self-supervised objectives, and then supervised trained with different amount of training labels. Different curves indicates different self-supervised objectives $\mathcal{L}$ and different $e(\mathbf{x})$ (denoted as "[]") indicated by the legend. The X axis indicates different training label ratio $\Gamma$ and the Y axis indicates the Top 1 accuracy from the joint predict of image classifier and $e(\mathbf{x})$. *No Prior* indicates an standalone image classifier with no location encoder. *Sup.* indicates training location encoders with supervised learning loss. The shaded area along each curve indicates the standard deviations we get after running each setting for 5 times. (a) Comparison among five models with identical model architecture but different training objectives $\mathcal{L}$ on iNat2018 dataset. MSE, $BI - \mathcal{L}_{BI}^{batch} + \mathcal{L}_{BI}^{negloc} + \mathcal{L}_{BI}^{simcse}$, and $MC - \mathcal{L}_{MC}^{batch} + \mathcal{L}_{MC}^{negloc} + \mathcal{L}_{MC}^{simcse}$ indicate training $sphereM+$ with three different self-supervised objectives before supervised training. (b) Comparing the performance of $sphereM+$ and $theory$ (the 2nd best model on iNat2018 dataset) in both supervised only (*Sup.*) and self-supervised plus supervised (*Sup.*&MC) settings. We can see that in both settings, $sphereM+$ is able to outperform $theory$ under different ratio $\Gamma$.

Figure 5a compares the model performance of $sphereM+$ on iNat2018 with different training objectives $\mathcal{L}$. Each curve indicates the model performance of one training process in different supervised label ratios $\Gamma$ with identical $e(\mathbf{x})$. From Figure 5a, we can see that compared with *Sup.* setting, all three self-supervised objectives can improve the model performances. The largest improvements happen when $\Gamma$ is small (i.e., few shot learning setting), and we **get a 3.9% performance improvement when adding** MC **self-supervised training stage for** $\Gamma = 0.05$. BI and MC show competitive performance while MC is slightly better. MSE is less effective than the two. Furthermore, in order to show the impact of self-supervised training on different $e(\mathbf{x})$, we compare the performance of $sphereM+$ and $theory$ (the 2nd best model on iNat2018 ) under different $\mathcal{L}$ (See Figure 5b). We can see that MC can improve the performance of both $e(\mathbf{x})$ while $sphereM+$ can outperform $theory$ in either supervised or self-supervised + supervised settings.

We also perform a series of ablation studies on these self-supervised training objectives. Please refer to Appendix 9.10 for detail analysis. Moreover, we also compare MC and *Sup.* on the fMoW dataset and find that MC self-supervised loss is also effective in the label limited learning setting on the fMoW dataset (more details in Figure 8 and Appendix 9.11).

## 6 CONCLUSION

In this work, we propose a multi-scale spherical location encoder - *Sphere2Vec* which can encode any location on the spherical surface into a high dimension vector. Moreover, we propose an self-supervised learning framework which pretrains location encoders based on unlabeled geo-tagged images. Experiment results on seven large-scale geo-aware image classification datasets shows that *Sphere2Vec* can outperform multiple strong baselines. Moreover, we show that by adding the self-supervised pretraining stage, the performance of *Sphere2Vec* can be further improved, especially when labeled data is limited. In the future, we would like to explore the effectiveness of *Sphere2Vec* on other machine learning tasks such as image geolocalization, geographic question answering, and so on. Moreover, we only pretrain the location encoder while keeping the image encoder freeze during the self-supervised training stage. In the future, we would like to explore ways to pretrain dual encoders at the same time or early fusion of image and location data.

## 7 ETHICS STATEMENT

All datasets utilized in this work are publicly available. We only do some data preprocessing such as image feature extraction. No new information is collected.

We find out that the geographic coverage of most datasets used in this work such as BirdSnap, NABird, iNat2017, iNat2018, YFCC, and fMoW are mostly concentrated on North American, Europe while they have poor geographic coverage on other regions and countries such as Russia, China, Africa, and so on (See Figure 4a). This kind of geographic bias will also affect the machine learning models trained on them as shown in Figure 4d. This paper shows the our *Sphere2Vec* is more robust than the state-of-the-art models for such bias. However, how to systematically handle such geographic bias should be an interesting research direction.

## 8 REPRODUCIBILITY STATEMENT

The proposed models as well as preprocessed datasets in this paper will be made public available through GitHub.

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

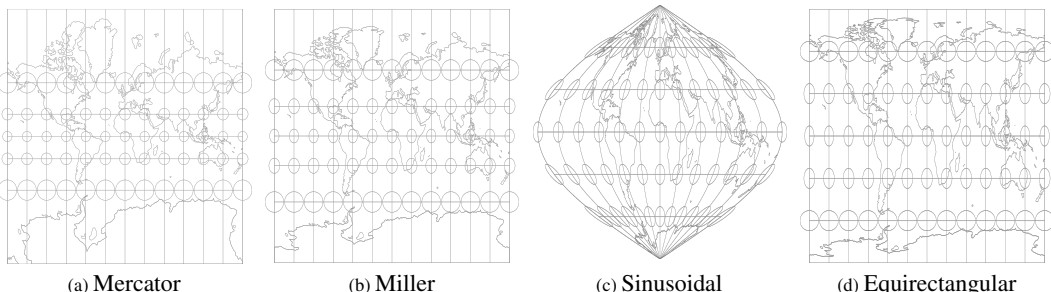

|  |  |  |  |
|---|---|---|---|
| (a) Mercator | (b) Miller | (c) Sinusoidal | (d) Equirectangular |

Figure 6: An illustration for map projection distortion: (a)-(d): Tissot indicatrices for four projections. The equal area circles are putted in different locations to show how the map distortion affect its shape.

# 9 APPENDIX

## 9.1 THE MAP PROJECTION DISTORTION PROBLEM

In fact there is no map projection, which can preserve distances in all directions. The so-called equidistant projection can only preserve distance on one direction, e.g., the longitude direction for the equirectangular projection (See Figure 6d), while the conformal map projections (See Figure 6a) can preserve directions while resulting in a large distance distortion. For a comprehensive overview of map projections and their distortions, see Mulcahy & Clarke (2001). This is a well recognized problem in Cartography which shows the importance of *calculating on a round planet* Chrisman (2017). Due to the limitations of these 2D location encoders, there is an urgent need for *a location encoding method which preserves the spherical distance (e.g., great circle distance[5]) between two points*. The multi-scale encoding method utilizes Double Fourier Sphere basis ($O(S^2)$ terms) or a subset ($O(S)$ terms) of it while still being able to correctly measure the spherical distance. This inspires us to explore the most effective subset of Fourier bases on spheres.

## 9.2 SPHEREC,SPHEREC+,SPHEREM,SPHEREM+

**sphereC**  Inspired by the fact that any point $(x, y, z)$ in 3D Cartesian coordinate can be expressed by $sin$ and $cos$ basis of spherical coordinates ($\lambda$, $\phi$ plus radius) [6], we define the basic form of *Sphere2Vec*, namely $sphereC$ encoder, for scale $s$ as

$$G^{(sphereC)}(\mathbf{x}) = \bigcup_{s=0}^{S-1} [\sin\phi_s, \cos\phi_s\cos\lambda_s, \cos\phi_s\sin\lambda_s]. \tag{9}$$

It can be shown that when $S = 1$, $G^{(sphereC)}(\mathbf{x})$ directly satisfies our expectation in Equation 1 where $f(x) = \cos(\frac{x}{R})$. See more detailed analysis and comparison to the $grid$ encoder in Appendix 9.3.

**sphereM**  Considering the fact that many geographical features are more sensitive to either latitude (e.g., temperature, sunshine duration) or longitude (e.g., timezones, geopolitical borderlines), we might want to focus on increasing the resolution of either $\phi$ or $\lambda$ while the other is hold relatively at large scale. Therefore, we introduce a multi-scale position encoder $sphereM$, where interaction terms between $\phi$ and $\lambda$ always have one of them fixed at top scale:

$$G^{(sphereM)}(\mathbf{x}) = \bigcup_{s=0}^{S-1} [\sin\phi_s, \cos\phi_s\cos\lambda, \cos\phi\cos\lambda_s, \cos\phi_s\sin\lambda, \cos\phi\sin\lambda_s]. \tag{10}$$

This new encoder ensures that the $\phi$ term interact with all the scales of $\lambda$ terms and $\lambda$ term interact with all the scales of $\phi$ terms. Note that when $S = 1$, $G^{(sphereM)}$ is equivalent to $G^{(sphereC)}$. Both $sphereC$ and $sphereM$ are multi-scale versions of a spherical distance-kept encoder (See Equation 12) and keep that as the main term in their multi-scale representation.

---

[5] https://en.wikipedia.org/wiki/Great-circle_distance
[6] https://en.wikipedia.org/wiki/Spherical_coordinate_system

**sphereC+ and sphereM+** From the analysis of the two proposed encoders and the state-of-the-art $grid$ encoders (Appendix 9.3), we know that $grid$ pays more attention to the sum of $cos$ difference of latitudes and longitudes, while our proposed encoders pay more attention to the spherical distances. In order to capture both information, we consider merging $grid$ with each proposed encoders to get more powerful models that encode geographical information from different angles.

$$G^{(sphereC+)}(\mathbf{x}) = G^{(sphereC)}(\mathbf{x}) \cup G^{(grid)}(\mathbf{x}),$$
$$G^{(sphereM+)}(\mathbf{x}) = G^{(sphereM)}(\mathbf{x}) \cup G^{(grid)}(\mathbf{x}). \tag{11}$$

### 9.3 ANALYSIS OF $sphereC$ AND ITS COMPARISON TO THE $grid$ ENCODER

To illustrate that $sphereC$ is good at capturing spherical distance, we take a close look at its basic case $S = 1$ (define $s = 0$ and $f_s = 1$), where the multi-scale encoder degenerates to

$$\widehat{G}^{(sphereC)}(\mathbf{x}) = [\sin(\phi), \cos(\phi)\cos(\lambda), \cos(\phi)\sin(\lambda)]. \tag{12}$$

These three terms are included in the multi-scale version ($S > 1$) and serve as the main terms at the largest scale and also the lowest frequency (when $s = S - 1$). The high frequency terms are added to help the downstream neuron network to learn the point-feature more efficiently. Interestingly, $\widehat{G}^{(sphereC)}$ captures the spherical distance in a very explicit way:

**Theorem 2.** *Let $\mathbf{x}_1$, $\mathbf{x}_2$ be two points on the same sphere with radius $R$, then*

$$\langle \widehat{G}^{(sphereC)}(\mathbf{x}_1), \widehat{G}^{(sphereC)}(\mathbf{x}_2) \rangle = \cos(\frac{\Delta D}{R}), \tag{13}$$

*where $\Delta D$ is the great circle distance between $\mathbf{x}_1$ and $\mathbf{x}_2$. Under this metric,*

$$\|\widehat{G}^{(sphereC)}(\mathbf{x}_1) - \widehat{G}^{(sphereC)}(\mathbf{x}_2)\| = 2\sin(\frac{\Delta D}{2R}). \tag{14}$$

*Moreover, $\|\widehat{G}^{(sphereC)}(\mathbf{x}_1) - \widehat{G}^{(sphereC)}(\mathbf{x}_2)\| \approx \frac{\Delta D}{R}$, when $\Delta D$ is small w.r.t. $R$.*

See the proof in Appendix 9.4. Since the central angle $\Delta\delta = \frac{\Delta D}{R} \in [0, \pi]$ and $\cos(x)$ is strictly monotonically decrease for $x \in [0, \pi]$, Theorem 2 shows that $\widehat{G}^{(sphereC)}(\mathbf{x})$ directly satisfies our expectation in Equation 1 where $f(x) = \cos(\frac{x}{R})$. In comparison, when $S = 1$, the inner product in the output space of $grid$ encoder is

$$\langle \widehat{G}^{(grid)}(\mathbf{x}_1), \widehat{G}^{(grid)}(\mathbf{x}_2) \rangle = \cos(\phi_1 - \phi_2) + \cos(\lambda_1 - \lambda_2), \tag{15}$$

which models the latitude difference and longitude difference of $\mathbf{x}_1$ and $\mathbf{x}_2$ separately rather than spherical distance. This introduces problems in encoding. For instance, consider data pairs $\mathbf{x}_1 = (\lambda_1, \phi)$ and $\mathbf{x}_2 = (\lambda_2, \phi)$, the distance between them in output space of $grid$, $\|\widehat{G}^{(grid)}(\mathbf{x}_1) - \widehat{G}^{(grid)}(\mathbf{x}_2)\|^2 = 2 - 2\cos(\lambda_1 - \lambda_2)$ stays as a constant in terms of $\phi$. However, when $\phi$ varies from $-\frac{\pi}{2}$ to $\frac{\pi}{2}$, the actual spherical distance changes in a wide range, e.g., the actual distance between the data pair at $\phi = -\frac{\pi}{2}$ (South Pole) is 0 while the distance between the data pair at $\phi = 0$ (Equator), gets the maximum value. This issue in measuring distances also has a negative impact on $grid$'s ability to model distributions in areas with sparse sample points because it is hard to learn the true spherical distance. We observe that $grid$ reaches peak performance at much smaller $r_{min}$ than that of *Sphere2Vec* encodings. Moreover, $sphereC$ outperforms $grid$ near polar regions where $grid$ claims large distance though the spherical distance is small (A, B in Figure 1).

### 9.4 PROOF OF THEOREM 2

*Proof.* Since $\widehat{G}^{(sphereC)}(\mathbf{x}_i) = [\sin(\phi_i), \cos(\phi_i)\cos(\lambda_i), \cos(\phi_i)\sin(\lambda_i)]$ for $i = 1, 2$, the inner product

$$
\begin{aligned}
&\langle \widehat{G}^{(sphereC)}(\mathbf{x}_1), \widehat{G}^{(sphereC)}(\mathbf{x}_2) \rangle \\
&= \sin(\phi_1)\sin(\phi_2) + \cos(\phi_1)\cos(\lambda_1)\cos(\phi_2)\cos(\lambda_2) + \cos(\phi_1)\sin(\lambda_1)\cos(\phi_2)\sin(\lambda_2) \\
&= \sin(\phi_1)\sin(\phi_2) + \cos(\phi_1)\cos(\phi_2)\cos(\lambda_1 - \lambda_2) \\
&= \cos(\Delta\delta) = \cos(\Delta D/R),
\end{aligned} \tag{16}
$$

where $\Delta\delta$ is the central angle between $\mathbf{x}_1$ and $\mathbf{x}_2$, and the spherical law of cosines is applied to derive the second last equality. So,

$$
\begin{aligned}
&\|\widehat{G}(\mathbf{x}_1) - \widehat{G}(\mathbf{x}_2)\|^2 \\
&= \langle \widehat{G}(\mathbf{x}_1) - \widehat{G}(\mathbf{x}_2), \widehat{G}(\mathbf{x}_1) - \widehat{G}(\mathbf{x}_2) \rangle \\
&= 2 - 2\cos(\Delta D/R) \\
&= 4\sin^2(\Delta D/2R).
\end{aligned}
\tag{17}
$$

So $\|\widehat{G}(\mathbf{x}_1) - \widehat{G}(\mathbf{x}_2)\| = 2\sin(\Delta D/2R)$ since $\Delta D/2R \in [0, \frac{\pi}{2}]$. By Taylor expansion, $\|\widehat{G}(\mathbf{x}_1) - \widehat{G}(\mathbf{x}_2)\| \approx \Delta D/R$ when $\Delta D$ is small w.r.t. $R$. □

### 9.5 Proof of Theorem 1

$\forall * \in \{sphereC, sphereC+, sphereM, sphereM+\}$, $G^{(*)}(\mathbf{x}_1) = G^{(*)}(\mathbf{x}_2)$ implies

$$
\sin(\phi_1) = \sin(\phi_2),
\tag{18}
$$

$$
\cos(\phi_1)\sin(\lambda_1) = \cos(\phi_2)\sin(\lambda_2),
\tag{19}
$$

$$
\cos(\phi_1)\cos(\lambda_1) = \cos(\phi_2)\cos(\lambda_2),
\tag{20}
$$

from $s = 0$ terms. Equation 18 implies $\phi_1 = \phi_2$. If $\phi_1 = \phi_2 = \pi/2$, then both points are at North Pole, $\lambda_1 = \lambda_2$ equal to whatever longitude defined at North Pole. If $\phi_1 = \phi_2 = -\pi/2$, it is similar case at South Pole. When $\phi_1 = \phi_2 \in (-\frac{\pi}{2}, \frac{\pi}{2})$, $\cos(\phi_1) = \cos(\phi_2) \neq 0$. Then from Equation 19 and 20,

$$
\sin\lambda_1 = \sin(\lambda_2), \cos(\lambda_1) = \cos(\lambda_2),
\tag{21}
$$

which shows that $\lambda_1 = \lambda_2$. In summary, $\mathbf{x}_1 = \mathbf{x}_2$, so $G^{(*)}$ is injective.
If $* = sphereDFS$, $G^{(*)}(\mathbf{x}_1) = G^{(*)}(\mathbf{x}_2)$ implies

$$
\sin(\phi_1) = \sin(\phi_2), \cos(\phi_1) = \cos(\phi_2), \sin(\lambda_1) = \sin(\lambda_2), \cos(\lambda_1) = \cos(\lambda_2),
\tag{22}
$$

which proves $\mathbf{x}_1 = \mathbf{x}_2$ and $G^{(*)}$ is injective directly.

### 9.6 Baselines

In order to understand the advantage of sphereical-distance-kept location encoders, we compare different versions of *Sphere2Vec* with multiple baselines:

- *No Prior* indicates a image classifier without using any location information, i.e., predicting image labels purely based on image information $P(y|\mathbf{I})$.

- *tile* divides the study area $A$ (e.g., the earth surface) into grids with equal intervals along the latitude and longitude direction. Each grid has an embedding to be used as the encoding for every location $\mathbf{x}$ fall into this grid. This is a common practice by many previous work when dealing with coordinate data (Berg et al., 2014; Adams et al., 2015; Tang et al., 2015).

- *wrap* is a location encoder model introduced by Mac Aodha et al. (2019). Given a location $\mathbf{x} = (\lambda, \phi)$, it uses a coordinate wrap mechanism to convert each dimension of $\mathbf{x}$ into 2 numbers - $G^{(wrap)}(\mathbf{x}) = [\sin(\pi\frac{\lambda}{180°}), \cos(\pi\frac{\lambda}{180°}), \sin(\pi\frac{\phi}{90°}), \cos(\pi\frac{\phi}{90°})]$. Then the results are passed through a multi-layered fully connected neural network $\mathbf{NN}^{(wrap)}()$ which consists of an initial fully connected layer, followed by a series of $h$ residual blocks, each consisting of two fully connected layers ($k$ hidden neurons) with a dropout layer in between. We adopt the official code of Mac Aodha et al. (2019)[7] for this implementation.

[7] http://www.vision.caltech.edu/~macaodha/projects/geopriors/

- $wrap + ffn$ is similar to $wrap$ except that it replaces $\mathbf{NN}^{(wrap)}()$ with $\mathbf{NN}_{ffn}()$, a simple learnable multi-layer perceptron with $h$ hidden layers and $k$ neurons per layer as that *Sphere2Vec* has. $wrap + ffn$ is used to exclude the effect of different $\mathbf{NN}()$ on the performance of location encoders. In the following, all location encoder baselines use $\mathbf{NN}_{ffn}()$ as the learnable neural network component so that we can directly compare the effect of different position encoding $G(\cdot)$ on the model performance.

- $xyz$ first converts $\mathbf{x}_i = (\lambda_i, \phi_i) \in \mathbb{S}^2$ into 3D Cartesian coordinates $(x, y, z)$ centered at the sphere center by following Equation 23 before feeding into a multilayer perceptron $\mathbf{NN}()$. Here, we let $(x, y, z)$ to locate on a unit sphere with radius $R = 1$. $xyz$ can be treated as a special case of $sphereC$ where $S = 1$.

$$G^{(xyz)}(\mathbf{x}) = [z, x, y] = [\sin\phi, \cos\phi\cos\lambda, \cos\phi\sin\lambda] \tag{23}$$

- $rbf$ randomly samples $M$ points from the training dataset as RBF anchor points $\{\mathbf{x}_m^{anchor}, m = 1...M\}$, and use gaussian kernels $\exp\left(-\dfrac{\|\mathbf{x}_i - \mathbf{x}_m^{anchor}\|^2}{2\sigma^2}\right)$ on each anchor points, where $\sigma$ is the kernel size. Each point $p_i$ has a $M$-dimension RBF feature vector which is fed into $\mathbf{NN}_{ffn}()$ to obtain the location embedding. This is a strong baseline for representing floating number features in machine learning models.

- $rff$, i.e., *Random Fourier Features* (Rahimi & Recht, 2008; Nguyen et al., 2017), first encodes location $\mathbf{x}$ into a $D$ dimension vector - $G^{(rff)}(\mathbf{x}) = \varphi(\mathbf{x}) = \dfrac{\sqrt{2}}{\sqrt{D}}[\cos(\omega_i^T\mathbf{x} + b_i)]_{i=1}^D$ where $\omega_i \overset{i.i.d}{\sim} \mathcal{N}(\mathbf{0}, \delta^2 I)$ and $b_i$ is uniformly sampled from $[0, 2\pi]$. $I$ is an identity matrix. Each component of $\varphi(\mathbf{x})$ first projects $\mathbf{x}$ into a random direction $\omega_i$ and makes a shift by $b_i$. Then it wraps this line onto the unit cirle in $\mathbb{R}^2$ with the cosine function. Rahimi & Recht (2008) show that $\varphi(\mathbf{x})^T\varphi(\mathbf{x}')$ is an unbiased estimate of the Gaussian kernal $K(\mathbf{x}, \mathbf{x}')$. $\varphi(\mathbf{x})$ is consist of $D$ different estimates to produce a further lower variance approximation. To make $rff$ comparable to other baselines, we feed $\varphi(\mathbf{x})$ into $\mathbf{NN}_{ffn}()$ to produce the final location embedding.

- $grid$ is a multi-scale location encoder on 2D Euclidean space proposed by Mai et al. (2020b). Here, we simply treat $\mathbf{x} = (\lambda, \phi)$ as 2D coordinate. It first use $G^{(grid)}(\mathbf{x})$ shown in Equation 3 to encode location $\mathbf{x}$ into multi-scale representation and then feed it into $\mathbf{NN}_{ffn}()$ to produce the final location embedding.

- $theory$ is another multi-scale location encoder on 2D Euclidean space proposed by Mai et al. (2020b). It use a position encoder $G^{(theory)}(\mathbf{x})$ shown in Equation 24.. Here, $\mathbf{x}^s = [\lambda_s, \phi_s] = [\frac{\lambda}{f_s}, \frac{\phi}{f_s}]$ and $\mathbf{a}_1 = [1, 0]^T, \mathbf{a}_2 = [-1/2, \sqrt{3}/2]^T, \mathbf{a}_3 = [-1/2, -\sqrt{3}/2]^T \in \mathbb{R}^2$ are three unit vectors which orient $2\pi/3$ apart from each other. The encoding results are feeɪnto $\mathbf{NN}_{ffn}()$ to produce the final location embedding.

$$G^{(theory)}(\mathbf{x}) = \bigcup_{s=0}^{S-1}\bigcup_{j=1}^{3}[\sin(\langle\mathbf{x}^s, \mathbf{a}_j\rangle), \cos(\langle\mathbf{x}^s, \mathbf{a}_j\rangle)]. \tag{24}$$

## 9.7 MODEL IMPLEMENTATION DETAIL

All types of *Sphere2Vec* as well as all baseline models we compared share the same model set up - $e^{(x)}(\mathbf{x}) = \mathbf{NN}_{ffn}(G^{(*)}(\mathbf{x}))$. Here, $G^{(*)}(\cdot)$ is a position encoder which preprocesses the input coordinates $\mathbf{x} = (\lambda, \phi) \in \mathbb{S}^2$ and outputs a position embedding $G^{(*)}(\mathbf{x})$. There are no learnable parameter in $G^{(*)}(\mathbf{x})$. The position encoders $G^{(*)}(\cdot)$ used for different *Sphere2Vec* are discussed in detail in Section 3.1 while $G^{(*)}(\cdot)$ of all baselines are discussed in Appendix 9.6. We can see that $G^{(*)}(\cdot)$ of $grid$, $theory$ and different types of *Sphere2Vec* encode the input coordinates in a multi-scale fashion by using different sinusoidal functions with different frequencies. Many previous work call this practice "Fourier input mapping" (Rahaman et al., 2019b; Tancik et al., 2020; Basri et al., 2020; Anokhin et al., 2021). The difference is that $grid$ and $theory$ use the Fourier features from 2D Euclidean space while our *Sphere2Vec* uses all or the subset of Double Fourier Sphere Features to take into account the sphereical geometry and the distance distortion it brings.

Except for $wrap$, $\mathbf{NN}_{ffn}(\cdot)$ used by all the other location encoders is a learnable multi-layer perceptron (MLP) with $h$ hidden layers and $k$ neurons per layer. We use the original MLP implementation provided by Mai et al. (2020b). Here, each layer of $\mathbf{NN}_{ffn}()$ is a linear layer followed by a dropout layer, a nonlinear activation function, a layer normalization layer (Ba et al., 2016), and a skip connection. Different from other location encoders, $wrap$ uses a different implementation of the full connected layer $\mathbf{NN}_{ffn}^{wrap}(\cdot)$. Please refer to Appendix 9.6 and Mac Aodha et al. (2019) for the detailed implementation of $\mathbf{NN}_{ffn}^{wrap}(\cdot)$.

All models are implemented in PyTorch. We use the original implementation of $wrap$ from Mac Aodha et al. (2019) and the implementation of $grid$ and $theory$ from Mai et al. (2020b). We train/evaluate each model on a Ubuntu machine with 2 GeForce GTX Nvidia GPU cores, each of which has 10GB memory.

## 9.8 MODEL TRAINING DETAIL AND HYPERPARAMETER TUNING

To train different types of *Sphere2Vec* as well as baseline models, we use Adam as the optimizer in both unsupervised and supervised training phase. Basically, we first do unsupervised training with an initial learning rate $\eta_{unsuper}$ by following either Equation 6 or 7 as the unsupervised training objective. In this setting, we predict the corresponding image embedding based on the input location. Only location encoder is trained during the unsupervised training phase while we keep the image encoder freezed as shown in Figure 1. Here, we just use a image encoder pretrained on other image datasets such as ImageNet and use it as a image feature extractor $\mathbb{F}$. After the model is convergence, we use a new learning rate $\eta_{super}$ to do supervised training by following Equation 8.

### 9.8.1 FULL SUPERVISED TRAINING SETTING

In order to test the effectiveness of our spherical encoding method – *Sphere2Vec*, we train all location encoders in a fully supervised setting. In order to find the best hyperparameter combinations for each model on each dataset, we use grid search to do hyperparameter tuning including supervised training learning rate $\eta_{super} = [0.01, 0.005, 0.002, 0.001, 0.0005, 0.00005]$, the number of scales $S = [16, 32, 64]$, the minimum scaling factor $r_{min} = [0.10.050.020.010.0050.0010.0001]$, the number of hidden layers and number of neurons used in $\mathbf{NN}_{ffn}(\cdot) - h = [1, 2, 3, 4]$ and $k = [256, 512, 1024]$, the dropout rate in $\mathbf{NN}_{ffn}(\cdot) - dropout = [0.1, 0.2, 0.3, 0.4, 0.5, 0.6, 0.7]$. We also test multiple options for the nonlinear function used for $\mathbf{NN}_{ffn}(\cdot)$ including ReLU, LeakyReLU, and Sigmoid. The maximum scaling factor $r_{max}$ can be determined based on the range of latitude $\phi$ and longitude $\lambda$. For $grid$ and $theory$, we use $r_{max} = 360$ and for all *Sphere2Vec*, we use $r_{max} = 1$. As for $rbf$ and $rff$, we also tune their hyperparamaters including kernel size $\sigma = [0.5, 1, 2, 10]$ as well as the number of kernels $M = [100, 200, 500]$.

Based on hyperparameter tuning, we find out using 0.5 as the dropout rate and ReLU as the nonlinear activation function for $\mathbf{NN}_{ffn}(\cdot)$ works best for every location encoder. Based on the hyperparameter tuning, the best hyperparameter combinations are selected for different models on different datasets. The best results are reported in Table 1. Note that each model has been running for 5 times and its mean Top1 score is reported. Due to the limit of space, the standard deviation of each model's performance on each dataset is not included in Table 1. However, we report the standard deviations of all models' performance on three datasets in Appendix 9.9.

Moreover, we find out $\eta_{super}$ and $r_{min}$ are the most important hyperparameters. Table 2 shows the best hyperparameter combinations of different *Sphere2Vec* models on different species image classification dataset. We use a smaller $S$ for $sphereDFS$ since it has $O(S^2)$ terms while the other models have $O(S)$ terms. $sphereDFS$ with $S = 8$ yield a similar number of terms to the other models wth $S = 32$ (See Table 3). Interestingly, all first four *Sphere2Vec* models ($sphereC$, $sphereC+$, $sphereM$, and $sphereM+$) shows the best performance on all five datasets with the same hyperparamter combinations. Note that compared with other datasets, iNat2017 and iNat2018 are more up-to-date datasets with more training samples and better geographic coverage. This indicates that the proposed 4 *Sphere2Vec* models show similar performance over different hyperparameter combinations.

Table 2: The best hyperparameter combinations of *Sphere2Vec* models on different image classification datasets. The learning rate $\eta_{super}$ tends to be smaller for larger datasets; The best $S$ is 8 for $sphereDFS$ and 32 for all others; and we fix the maximum scale $r_{max}$ as 1. Here, $r_{min}$ indicates the minimum scale. $h$ and $k$ are the number of hidden layers and the number of neurons in $\mathbf{NN}()$ respectively.

| Dataset | $\eta_{super}$ | $r_{min}$ | $k$ |
|---|---|---|---|
| BirdSnap | 0.001 | $10^{-6}$ | 512 |
| BirdSnap† | 0.001 | $10^{-4}$ | 1024 |
| NABirds† | 0.001 | $10^{-4}$ | 1024 |
| iNat2017 | 0.0001 | $10^{-2}$ | 1024 |
| iNat2018 | 0.0005 | $10^{-3}$ | 1024 |

Table 3: Dimension of position encoding for different models in terms of total scales $S$

| Model | $sphereC$ | $sphereC+$ | $sphereM$ | $sphereM+$ | $sphereDFS$ |
|---|---|---|---|---|---|
| Dimension | $3S$ | $6S$ | $5S$ | $8S$ | $4S^2 + 4S$ |

We also find out that using a deeper MLP as $\mathbf{NN}_{ffn}(\cdot)$, i.e., a larger $h$ does not necessarily lead to better classification accuracy. In many cases, one hidden layer – $h = 1$ achieves the best performance for many kinds of location encoders. We discuss this in detail in Appendix 9.9.

### 9.8.2 UNSUPERVISED TRAINING SETTING

In order to test the effectiveness of different unsupervised learning objectives, we also conduct another set of experiments by unsupervised training different location encoders before the supervised training phase. In addition to the hyperparameter we discussed in Section 9.8.1, we also fine tune the hyperparameters used in the unsupervised learning phase including the unsupervised training learning rate $\eta_{unsuper}$, the negative location loss weight $\alpha_1$, SimCSE loss weight $\alpha_2$, the number of negative locations we used $|\mathcal{N}^s|$, three temperature $\tau_0$, $\tau_1$, and $\tau_2$. As for unsupervised objective $\mathcal{L}_{BI}$, we also fine tune $\beta_1$ and $\beta_2$. The hyperparameters we have searched and the hyperparameter tuning results for the unsupervised learning setting are shown in Table 5, 6, 7 in Appedix 9.10.

### 9.9 MORE EXPERIMENTAL RESULTS ON GEO-AWARE IMAGE CLASSIFICATION TASKS

Table 4 is a complementary of Table 1 which compares the performance of $sphereM+$ with all baseline models on the geo-aware image classification task. The results on three datasets are shown here including BirdSnap†, NABirds†, and iNat2018. For each model, we vary the depth of its $\mathbf{NN}_{ffn}()$, i.e., $h = [1, 2, 3, 4]$. The best evaluation results with each $h$ are reported. Moreover, we run each model with one specific $h$ 5 times and report the standard deviation of the Top1 accuray, indicated in "()". Several important observations can be made based on Table 4:

1. Although the absolute performance improvement between $sphereM+$ and the best baseline model is not very large – 0.90%, 0.28%, and 0.71% for three datasets respectively, **these performance improvements are statistically significant given the standard deviations of these Top1 scores.**

2. **These performance improvements are comparable to those from the previous studies on the same tasks.** In other words, the small margin is due to the nature of these datasets. For example, Mai et al. (2020b) showed that $grid$ or $theory$ has 0.79%, 0.44% absolute Top1 accuracy improvement on BirdSnap† and NABirds† dataset respectively. Mac Aodha et al. (2019) showed that $wrap$ has 0.09%, 0%, 0.04% absolute Top1 accuracy improvement on BirdSnap, BirdSnap† and NABirds† dataset. Here, we only consider the results of $wrap$ that only uses location information, but not temporal information. Although Mac Aodha et al. (2019) showed that compared with $tile$ and nearest neighbor methods, $wrap$ has 3.19% and 3.71% performance improvement on iNat2017 and iNat2018 datatset, these large margins are mainly because the baselines they used are rather weak. When we consider the typical $rbf$ and $rff$ (Rahimi et al., 2007) used in our study, their performance improvement are down to -0.02% and 0.61%.

Table 4: The impact of the depth $h$ of multi-layer perceptrons $\mathbf{NN}_{ffn}()$ on Top1 accuracy. The numbers in "()" indicates the standard deviations estimated from 5 independent train/test runs. We find that the model performances are not very sensitive to $\mathbf{NN}_{ffn}()$, and, in most cases, one layer $\mathbf{NN}_{ffn}()$ achieve the best result. In other words, the larger performance gaps in fact comes from different $G^{(*)}(\cdot)$ we use. Moreover, given the performance's variance of each model, we can see that $sphereM+$ outperforms other baseline models on all these three datasets and the margins are statistically significant. The same conclusion can be drawn based on our experiments on other datasets. Here, we only show results on three datasets as an illustrative example.

| Dataset | | BirdSnap† | NABirds† | iNat2018 |
|---|---|---|---|---|
| P(y\|x) - Prior Type | $h$ | Test | Test | Val |
| | 1 | 78.81 (0.10) | 81.08 (0.05) | 71.6 (0.08) |
| $xyz$ | 2 | 78.83 (0.10) | 81.2 (0.09) | 71.7 (0.02) |
| | 3 | 78.97 (0.06) | 81.11 (0.06) | 71.75 (0.04) |
| | 4 | 78.84 (0.09) | 81.02 (0.03) | 71.71 (0.03) |
| | 1 | 79.04 (0.13) | 81.60 (0.04) | 72.89 (0.08) |
| $wrap$ (Mac Aodha et al., 2019) | 2 | 78.94 (0.13) | **81.62 (0.04)** | 72.84 (0.07) |
| | 3 | 79.08 (0.15) | 81.53 (0.02) | 72.92 (0.05) |
| | 4 | 79.06 (0.11) | 81.51 (0.09) | 72.77 (0.06) |
| | 1 | 78.97 (0.09) | 81.23 (0.06) | 71.90 (0.05) |
| $wrap + ffn$ | 2 | 79.02 (0.15) | 81.36 (0.04) | 71.95 (0.05) |
| | 3 | 79.21 (0.14) | 81.35 (0.05) | 71.94 (0.04) |
| | 4 | 79.06 (0.09) | 81.27 (0.13) | 71.93 (0.04) |
| | 1 | 79.40 (0.13) | 81.32 (0.08) | 71.02 (0.18) |
| $rbf$(Mai et al., 2020b) | 2 | 79.38 (0.12) | 81.22 (0.11) | 71.29 (0.20) |
| | 3 | 79.40 (0.04) | 81.31 (0.07) | 71.35 (0.21) |
| | 4 | 79.25 (0.05) | 81.30 (0.07) | 71.21 (0.19) |
| | 1 | 78.96 (0.18) | 81.27 (0.07) | 71.76 (0.06) |
| $rff$(Rahimi et al., 2007) | 2 | 78.97 (0.04) | 81.28 (0.05) | 71.71 (0.09) |
| | 3 | 79.07 (0.12) | 81.30 (0.11) | 71.80 (0.04) |
| | 4 | 79.16 (0.13) | 81.22 (0.11) | 71.46 (0.05) |
| | 1 | **79.44 (0.11)** | 81.24 (0.06) | 72.98 (0.05) |
| $grid$ (Mai et al., 2020b) | 2 | 79.07 (0.05) | 81.13 (0.04) | 72.9 (0.03) |
| | 3 | 79.27 (0.10) | 81.04 (0.05) | 72.72 (0.04) |
| | 4 | 78.88 (0.02) | 80.78 (0.07) | 72.49 (0.03) |
| | 1 | 79.24 (0.15) | 81.23 (0.02) | **73.01 (0.09)** |
| $theory$ (Mai et al., 2020b) | 2 | 79.16 (0.21) | 81.31 (0.14) | 72.70 (0.02) |
| | 3 | 78.89 (0.21) | 81.07 (0.01) | 72.57 (0.07) |
| | 4 | 79.06 (0.18) | 80.72 (0.12) | 72.33 (0.08) |
| | 1 | **80.34 (0.08)** | 81.86 (0.03) | 73.72 (0.07) |
| $sphereM+$ | 2 | 79.91 (0.12) | 81.84 (0.04) | 73.46 (0.03) |
| | 3 | 80.06 (0.06) | **81.90 (0.05)** | **73.73 (0.02)** |
| | 4 | 79.92 (0.14) | 81.84 (0.10) | 73.21 (0.04) |

3. By comparing the performances of the same model with different depth of its $\mathbf{NN}_{ffn}()$, i.e., $h$, we can see that the model performance is not sensitive to $h$. In fact, in most cases, one layer $\mathbf{NN}_{ffn}()$ achieve the best result. This indicates that **the depth of the MLP does not significantly affect the model performance and a deeper MLP does not necessarily lead to a better performance**. In other words, **the systematic bias (i.e., distance distortion) introduced by** $grid$ **and** $theory$ **can not later be compensated by a deep MLP.** It shows the importance of designing a spherical-distance-aware location encoder.

## 9.10 ABLATION STUDY ON DIFFERENT SELF-SUPERVISED LOSS

Both the BI (Equation 6) and MC (Equation 7) unsupervised loss have three loss component: in-batch, negative location, SimCSE loss component. How does each of them contribute to the overall unsupervised training? In order to answering this question, we do ablation studies on these two

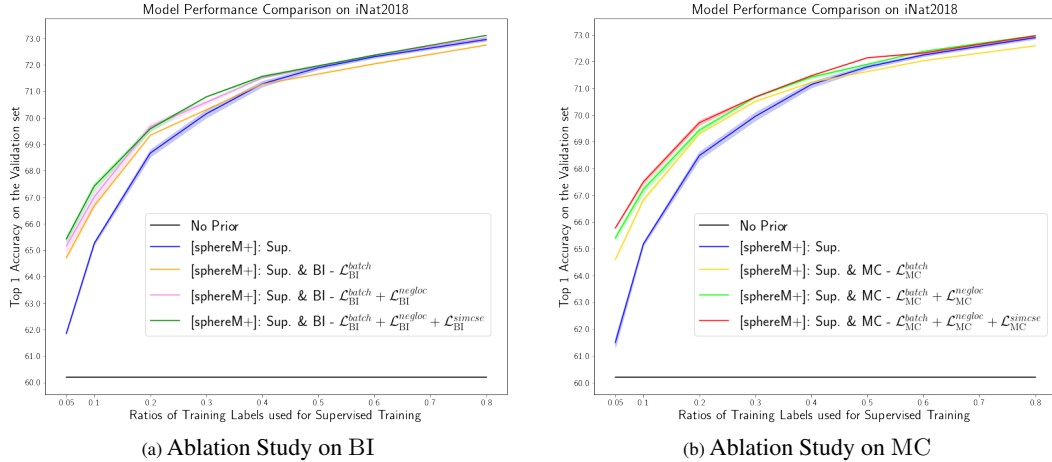

(a) Ablation Study on BI        (b) Ablation Study on MC

Figure 7: Ablation study on different unsupervised learning setting on iNat2018 dataset. (a) Alblation study on BI loss. $BI - \mathcal{L}_{BI}^{batch} + \mathcal{L}_{BI}^{negloc} + \mathcal{L}_{BI}^{simcse}$ indicates the full BI loss while $BI - \mathcal{L}_{BI}^{batch} + \mathcal{L}_{BI}^{negloc}$ deletes the SimCSE component. $BI - \mathcal{L}_{BI}^{batch}$ additionally deletes the negative location loss component. Comparing among those three model settings can help us understand the effect of different loss components. Similarly, we do the same ablation study on MC loss and show in (b).

objectives by using iNat2018 dataset as an example. Figure 7a and 7b illustrate the ablation study results.

From Figure 7a, we see that as for BI, adding $\mathcal{L}_{BI}^{negloc}$ will significantly increase the model performance especially when $\Gamma$ is large. Adding $\mathcal{L}_{BI}^{simcse}$ also improves the model performance when $\Gamma$ is small. As for the ablation study on MC loss as shown in Figure 7b, adding $\mathcal{L}_{MC}^{negloc}$ can significantly increase the model performance and adding $\mathcal{L}_{MC}^{simcse}$ can also make the performance slightly better.

## 9.11 EFFECTIVENESS OF UNSUPERVISED PRETRAINING ON FMOW DATASET

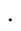

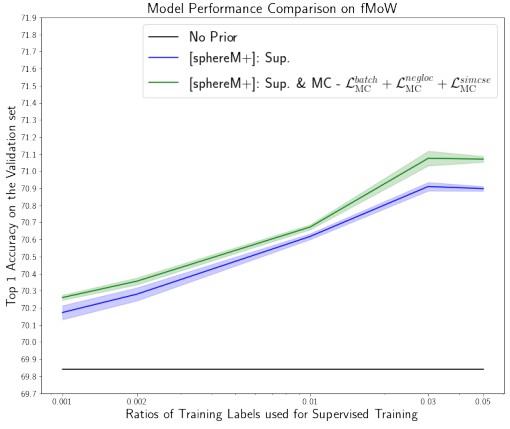

Figure 8: Comparison among two models with identical model architecture but different training objectives on fMoW dataset. Each point on each curve indicates a specific training process. We repeat each of them for five times and show the standard deviations as shaded areas along the line. We can see that with statistically significance, MC unsupervised loss is also effective in the few shot learning setting on the fMoW dataset.

## 9.12 IMPACT OF MRR BY THE NUMBER OF SAMPLES AT DIFFERENT LATITUDE BANDS

See Figure 9

Table 5: Ablation Study on unsupervised loss $\mathcal{L}_{\mathrm{MC}}$ over iNat2018 dataset. We show the effect of different hyperparemeters of $\mathcal{L}_{\mathrm{MC}}$ on the performance of location encoders. We use $sphereM+$ as an representative location encoder and use supervised training dataset ratio $\Gamma$ as 0.5. $lr$ indicates the learning rate used for unsupervised $\mathcal{L}_{\mathrm{MC}}$ training. $dropout$ indicates the dropout rate used by location encoder which will affect the SimCSE loss as Gao et al. (2021) shows. $\alpha_1$ and $\alpha_2$ are weights for the negative location loss $\mathcal{L}_{\mathrm{MC}}^{negloc}(\mathbb{X})$ and SImCSE loss $\mathcal{L}_{\mathrm{MC}}^{simcse}(\mathbb{X})$ component respectively. $\tau_0$, $\tau_1$, and $\tau_2$ are the temperatures used by three different loss components. See Equation 7 for the explanation.

| $\Gamma$ | $e(\mathbf{x})$ | $\mathcal{L}$ | $\eta_{unsuper}$ | $\alpha_1$ | $|\mathcal{N}^s|$ | $\alpha_2$ | $\tau_0$ | $\tau_1$ | $\tau_2$ | $dropout$ | Top1 |
|---|---|---|---|---|---|---|---|---|---|---|---|
| 0.5 | $sphereM+$ | $\mathcal{L}_{\mathrm{MC}}$ | 0.001 | 1 | 1 | 1 | 1 | 1 | 1 | 0.5 | 71.31 |
| | | | 0.0005 | 1 | 1 | 1 | 1 | 1 | 1 | 0.5 | **71.86** |
| | | | 0.0002 | 1 | 1 | 1 | 1 | 1 | 1 | 0.5 | 71.78 |
| | | | 0.00005 | 1 | 1 | 1 | 1 | 1 | 1 | 0.5 | 71.76 |
| | | | 0.00001 | 1 | 1 | 1 | 1 | 1 | 1 | 0.5 | 71.83 |
| | | | 0.000005 | 1 | 1 | 1 | 1 | 1 | 1 | 0.5 | 71.7 |
| | | | 0.000001 | 1 | 1 | 1 | 1 | 1 | 1 | 0.5 | 71.69 |
| | | | 0.0005 | 2 | 1 | 1 | 1 | 1 | 1 | 0.5 | 71.8 |
| | | | 0.0005 | 1 | 1 | 1 | 1 | 1 | 1 | 0.5 | **71.86** |
| | | | 0.0005 | 0.5 | 1 | 1 | 1 | 1 | 1 | 0.5 | 71.71 |
| | | | 0.0005 | 0 | 1 | 1 | 1 | 1 | 1 | 0.5 | 71.42 |
| | | | 0.0005 | 1 | 1 | 2 | 1 | 1 | 1 | 0.5 | 71.71 |
| | | | 0.0005 | 1 | 1 | 1 | 1 | 1 | 1 | 0.5 | 71.86 |
| | | | 0.0005 | 1 | 1 | 0.5 | 1 | 1 | 1 | 0.5 | 71.81 |
| | | | 0.0005 | 1 | 1 | 0 | 1 | 1 | 1 | 0.5 | 71.82 |
| | | | 0.0005 | 1 | 1 | 1 | 64 | 1 | 1 | 0.5 | 71.85 |
| | | | 0.0005 | 1 | 1 | 1 | 32 | 1 | 1 | 0.5 | 71.95 |
| | | | 0.0005 | 1 | 1 | 1 | 24 | 1 | 1 | 0.5 | 71.74 |
| | | | 0.0005 | 1 | 1 | 1 | 22 | 1 | 1 | 0.5 | 71.81 |
| | | | 0.0005 | 1 | 1 | 1 | 20 | 1 | 1 | 0.5 | **72.08** |
| | | | 0.0005 | 1 | 1 | 1 | 18 | 1 | 1 | 0.5 | 71.77 |
| | | | 0.0005 | 1 | 1 | 1 | 16 | 1 | 1 | 0.5 | 72.02 |
| | | | 0.0005 | 1 | 1 | 1 | 12 | 1 | 1 | 0.5 | 71.87 |
| | | | 0.0005 | 1 | 1 | 1 | 8 | 1 | 1 | 0.5 | 71.93 |
| | | | 0.0005 | 1 | 1 | 1 | 4 | 1 | 1 | 0.5 | 71.69 |
| | | | 0.0005 | 1 | 1 | 1 | 2 | 1 | 1 | 0.5 | 71.89 |
| | | | 0.0005 | 1 | 1 | 1 | 1 | 1 | 1 | 0.5 | 71.86 |
| | | | 0.0005 | 1 | 1 | 1 | 0.1 | 1 | 1 | 0.5 | 71.67 |
| | | | 0.0005 | 1 | 1 | 1 | 0.01 | 1 | 1 | 0.5 | 71.63 |
| | | | 0.0005 | 1 | 1 | 1 | 20 | 2 | 1 | 0.5 | 71.85 |
| | | | 0.0005 | 1 | 1 | 1 | 20 | 1 | 1 | 0.5 | **72.08** |
| | | | 0.0005 | 1 | 1 | 1 | 20 | 0.5 | 1 | 0.5 | 71.76 |
| | | | 0.0005 | 1 | 1 | 1 | 20 | 1 | 2 | 0.5 | 71.84 |
| | | | 0.0005 | 1 | 1 | 1 | 20 | 1 | 1 | 0.5 | **72.08** |
| | | | 0.0005 | 1 | 1 | 1 | 20 | 1 | 0.5 | 0.5 | 71.69 |
| | | | 0.0005 | 1 | 1 | 1 | 20 | 1 | 1 | 0.5 | **72.08** |
| | | | 0.0005 | 1 | 4 | 1 | 20 | 1 | 1 | 0.5 | 71.84 |
| | | | 0.0005 | 1 | 8 | 1 | 20 | 1 | 1 | 0.5 | 71.69 |
| | | | 0.0005 | 1 | 1 | 1 | 20 | 1 | 1 | 0.7 | 71.38 |
| | | | 0.0005 | 1 | 1 | 1 | 20 | 1 | 1 | 0.6 | 71.91 |
| | | | 0.0005 | 1 | 1 | 1 | 20 | 1 | 1 | 0.5 | **72.08** |
| | | | 0.0005 | 1 | 1 | 1 | 20 | 1 | 1 | 0.4 | 71.55 |
| | | | 0.0005 | 1 | 1 | 1 | 20 | 1 | 1 | 0.3 | 71.08 |
| | | | 0.0005 | 1 | 1 | 1 | 20 | 1 | 1 | 0.2 | 69.96 |
| | | | 0.0005 | 1 | 1 | 1 | 20 | 1 | 1 | 0.1 | 68.17. |

Table 6: Ablation Study on unsupervised loss $\mathcal{L}_{\mathrm{BI}}$ over iNat2018 dataset. We show the effect of different hyperparemeters of $\mathcal{L}_{\mathrm{BI}}$ on the performance of location encoders. We use $sphereM+$ as an representative location encoder ad use supervised training dataset ratio $\Gamma$ as 0.5. $lr$ indicates the learning rate used for unsupervised $\mathcal{L}_{\mathrm{MC}}$ training.

| $\Gamma$ | $e(\mathbf{x})$ | $\mathcal{L}$ | $\eta_{unsuper}$ | $\beta_1$ | $|\mathcal{N}^s|$ | $\beta_2$ | dropout | Top1 |
|---|---|---|---|---|---|---|---|---|
| 0.5 | $sphereM+$ | $\mathcal{L}_{\mathrm{BI}}$ | 0.001 | 1 | 1 | 0 | 0.5 | 71.76 |
| | | | 0.0008 | | | | | 71.87 |
| | | | 0.0005 | | | | | 71.85 |
| | | | 0.0002 | | | | | **72.02** |
| | | | 0.0001 | | | | | 71.67 |
| | | | 0.00001 | | | | | 71.62 |
| | | | 0.000005 | | | | | 71.48 |
| | | | 0.000002 | | | | | 71.56 |
| | | | 0.0002 | 2 | 1 | 1 | 0.5 | 71.98 |
| | | | | 1 | | | | **72.02** |
| | | | | 0.5 | | | | 71.85 |
| | | | | 0 | | | | 71.65 |
| | | | 0.0002 | 1 | 1 | | 0.5 | **72.02** |
| | | | | | 4 | | | 71.82 |
| | | | | | 8 | | | 71.94 |
| | | | | | 16 | | | 71.84 |
| | | | 0.0002 | 1 | 1 | 0.00 | 0.5 | **72.02** |
| | | | | | | 0.10 | | 72.06 |
| | | | | | | 0.20 | | 71.93 |
| | | | | | | 0.50 | | 71.83 |
| | | | | | | 1.00 | | 71.84. |

Table 7: Ablation Study on unsupervised loss $MSE$ over iNat2018 dataset. We show the effect of different hyperparemeters of $MSE$ on the performance of location encoders. We use $sphereM+$ as an representative location encoder ad use supervised training dataset ratio $\Gamma$ as 0.5. $lr$ indicates the learning rate used for unsupervised $MSE$ training.

| $\Gamma$ | $e(\mathbf{x})$ | $\mathcal{L}$ | $\eta_{unsuper}$ | dropout | Top1 |
|---|---|---|---|---|---|
| 0.5 | $sphereM+$ | MSE | 0.001 | 0.5 | 71.05 |
| | | | 0.0005 | | 71.41 |
| | | | 0.0001 | | 71.55 |
| | | | 0.00001 | | 71.58 |
| | | | 0.000005 | | 71.69 |
| | | | 0.000002 | | **71.71** |
| | | | 0.000001 | | 71.6 |

(a) iNat2017

(b) iNat2018

Figure 9: Impact of MRR by The Number of Samples at Different Latitude Bands.

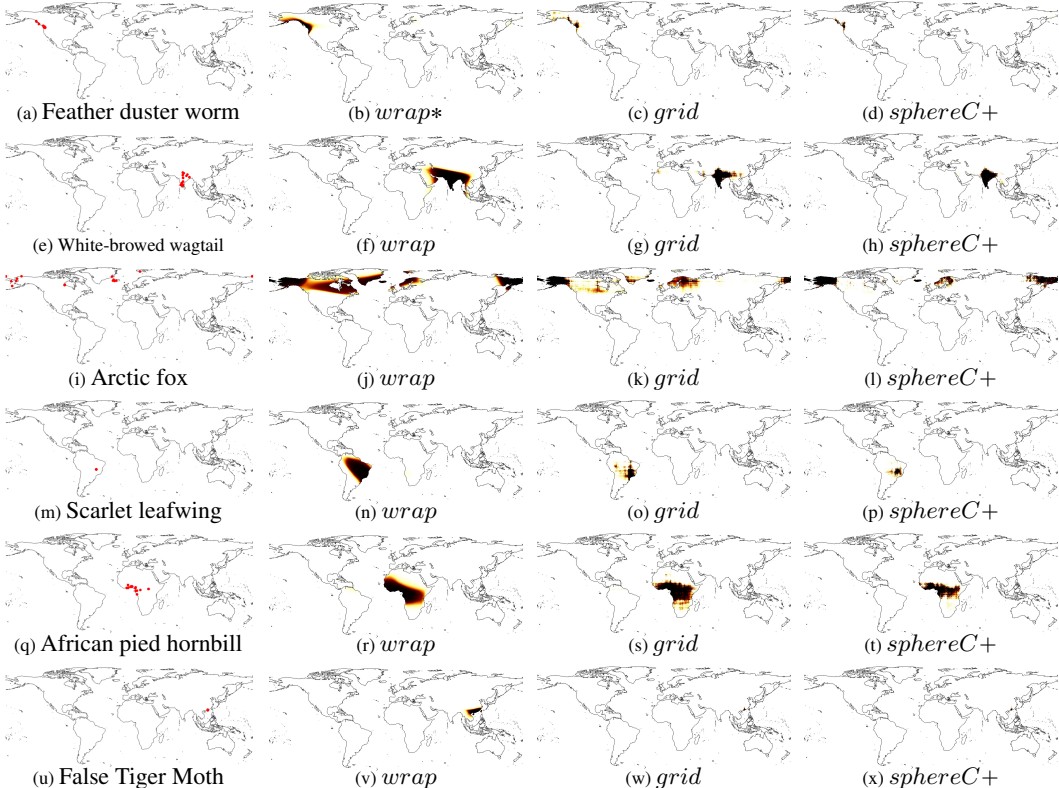

Figure 10: Compare the predicted distributions of example species from different models. The first figure of each row marks the data points from iNat2018 training data.

## 9.13 PREDICTED DISTRIBUTIONS INAT2018

We plot the predicted species distributions from different models at different geographic regions, and compare them with the training sample locations of the corresponding species, see Figure 10. We can see that compared with $wrap*$ and $grid$, in each geographic region with sparse training samples and the North Pole area, the spatial distributions produced by $sphereC+$ are more compact while the other two have over-generalization issue.

## 9.14 EMBEDDING CLUSTERING

We use the location encoder trained on iNat2017 or iNat2018 dataset to produce a location embedding for the center of each small latitude-longitude cell. Then we do agglomerative clustering[8] on all these embeddings to produce a clustering map. Figure 11 and 12 show the clustering results for different models with different hyperparameters on iNat2017 and iNat2018 dataset.

---

[8] https://scikit-learn.org/stable/modules/generated/sklearn.cluster.AgglomerativeClustering.html

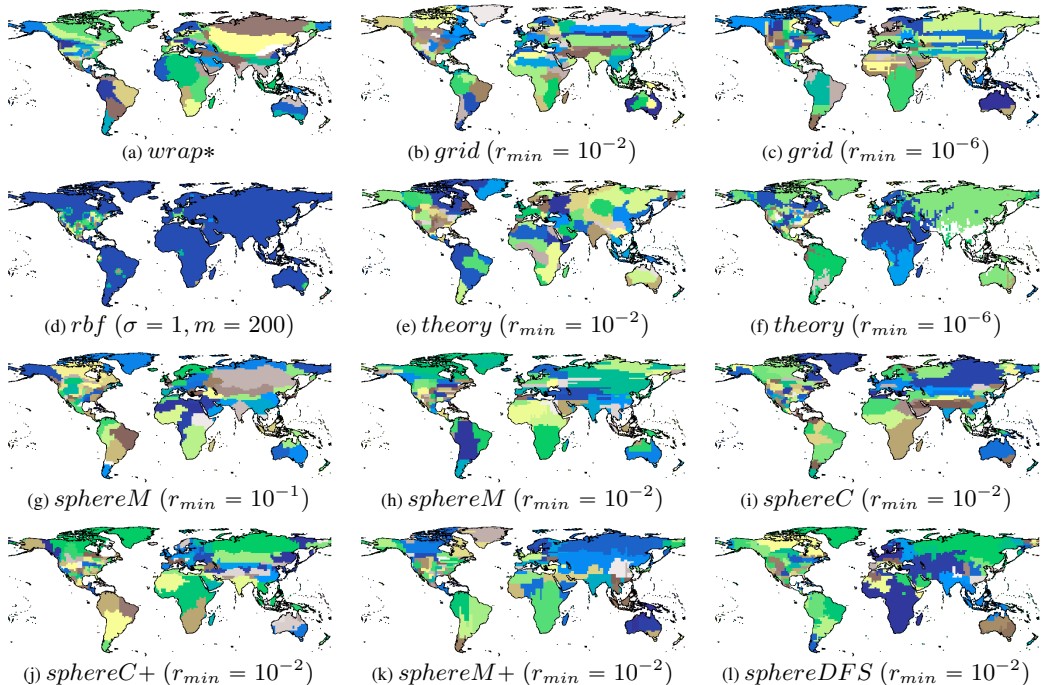

Figure 11: Embedding clusterings of iNat2017 models. (a) $wrap*$ with 4 hidden ReLU layers of 256 neurons; (d) $rbf$ with the best kernel size $\sigma = 1$ and number of anchor points $m = 200$; (b)(c)(e)(f) are *Space2Vec* models Mai et al. (2020b) with different min scale $r_{min} = \{10^{-6}, 10^{-2}\}$.[a] (g)-(l) are different *Sphere2Vec* models.[b]

[a] They share the same best hyperparameters: $S = 64$, $r_{max} = 1$, and 1 hidden ReLU layers of 512 neurons.
[b] They share the same best hyperparameters: $S = 32$, $r_{max} = 1$, and 1 hidden ReLU layers of 1024 neurons.

## 9.15    EXPERIMENTS WITH SYNTHETIC DATASET

### 9.15.1    SYNTHETIC DATASET GENERATION

To further investigate the effectiveness of our proposed spherical location encoder *Sphere2Vec*, we construct a set of synthetic datasets.

We utilize the von Mises–Fisher distribution ($vMF$) (Izbicki et al., 2019), an analogy of the 2D Gaussian distribution on the spherical surface $\mathbb{S}^2$ to generate synthetic data points[9]. The probability density function of $vMF$ is defined as

$$vMF(\mathbf{x}; \mu, \kappa) = \frac{\kappa}{2\pi \sinh(\kappa)} \exp(\kappa \mu^T \chi(\mathbf{x}))$$
$$where \; \chi(\mathbf{x}) = [x, y, z] = [\cos\phi\cos\lambda, \cos\phi\sin\lambda, \sin\phi] \tag{25}$$

Here, $\chi(\mathbf{x})$ converts $\mathbf{x}$ into a coordinates in the 3D Euclidean space, and on the surface of a unit sphere. A $vMF$ distribution is controlled by two parameters – the mean direction $\mu \in \mathbb{R}^3$ and concentration parameter $\kappa \in \mathbb{R}^+$. $\mu$ indicates the center of a $vMF$ distribution which is a 3D unit vector. $\kappa$ is a positive real number which controls the concentration of $vMF$. A higher $\kappa$ indicates more compact $vMF$ distribution, while $\kappa = 1$ correspond to a $vMF$ distribution with standard deviation covering half of the unit sphere.

To simulate multi-modal distributions, we assume $\mathcal{C}$ classes with even prior, and each classes follows a $vMF$ distribution. To create a dataset we first sample $\mathcal{C}$ sets of parameters $\{(\mu_i, \kappa_i)\}$ ($\mathcal{C} = 50$). Then we draw $\mathcal{SP}$ samples for each class ($\mathcal{SP} = 100$). So in total, each generated synthetic dataset has 5000 data points for 50 balanced classes.

---

[9] https://www.tensorflow.org/probability/api_docs/python/tfp/distributions/VonMisesFisher#sample

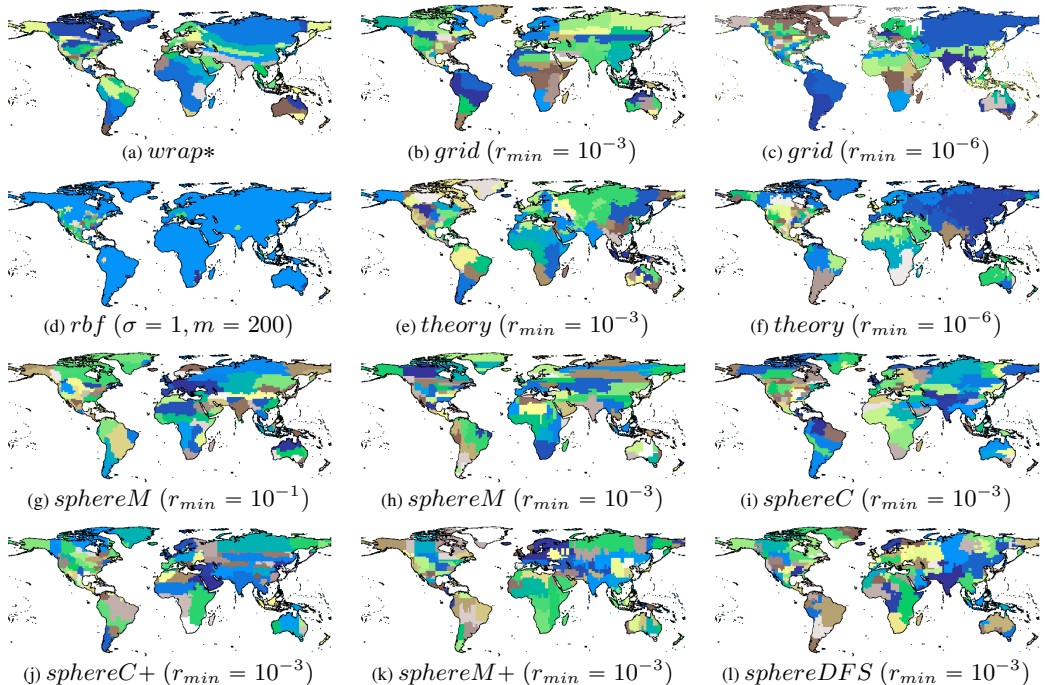

Figure 12: Embedding clusterings of iNat2018 models. (a) $wrap*$ with 4 hidden ReLU layers of 256 neurons; (d) $rbf$ with the best kernel size $\sigma = 1$ and number of anchor points $m = 200$; (b)(c)(e)(f) are *Space2Vec* models (Mai et al., 2020b) with different min scale $r_{min} = \{10^{-6}, 10^{-3}\}$.[a] (g)-(l) are *Sphere2Vec* models with different min scale.[b]

[a] They share the same best hyperparameters: $S = 64$, $r_{max} = 1$, and 1 hidden ReLU layers of 512 neurons.
[b] They share the same best hyperparameters: $S = 32$, $r_{max} = 1$, and 1 hidden ReLU layers of 1024 neurons.

The concentration parameter $\kappa_i$ is sampled by first drawing $r$ from an uniform distribtuion $U(\kappa_{min}, \kappa_{max})$, and then take the square $r^2$. The square helps to avoid sampling many large $\kappa_i$ which yield very concentrated $vMF$ distributions that are rather easy to be classified. We fix $\kappa_{min} = 1$ and vary $\kappa_{max}$ in $[16, 32, 64, 128]$.

For the center parameter $\mu_i$ we adopt two sampling approaches:

1. **Uniform Sampling**: We uniformly sample $\mathcal{C}$ centers ($\mu_i$) from the surface of a unit sphere. We generate four synthetic datasets (for different values of $\kappa_{max}$) and indicate them as U1, U2, U3, U4. See Table 8 for the parameters we use to generate these datasets.

2. **Stratified Sampling**: We first evenly divide the latitude range $[-\pi/2, \pi/2]$ into $N_\mu$ intervals. Then we uniformly sample $\mathcal{C}_\mu$ centers ($\mu_i$) from the spherical surface defined by each latitude interval. Since the latitude intervals in polar regions have smaller spherical surface area, this stratified sampling method has higher density in the polar regions. We keep $N_\mu \times \mathcal{C}_\mu = \mathcal{C} = 50$ fixed and varies $N_\mu$ in $[5, 10, 25, 50]$. Combined with the 4 $\kappa_{max}$ choices, this procedure yields 16 different synthetic datasets. We denote them as $Si.j$. See Table 8 for the parameters we use to generate these datasets.

Figure 13a-13d visualize the data point distributions of these four synthetic datasets based on the uniform sampling method in 2D space. Figure 13e visualized the U4 dataset in a 3D Euclidean space. We can see that when $\kappa_{max}$ is larger, the variation of point density among different $vMF$ distributions becomes larger. Some $vMF$ are very concentrated and the resulting data points are easier to be classified. Moreover, if we treat these datasets as 2D data points as $grid$ and $theory$ do, $vMF$ distributions in the polar areas will be stretched to very extended shapes making model learning more difficult. However, this kind of systematic bias can be avoided if we use a spherical location encoder as *Sphere2Vec*.

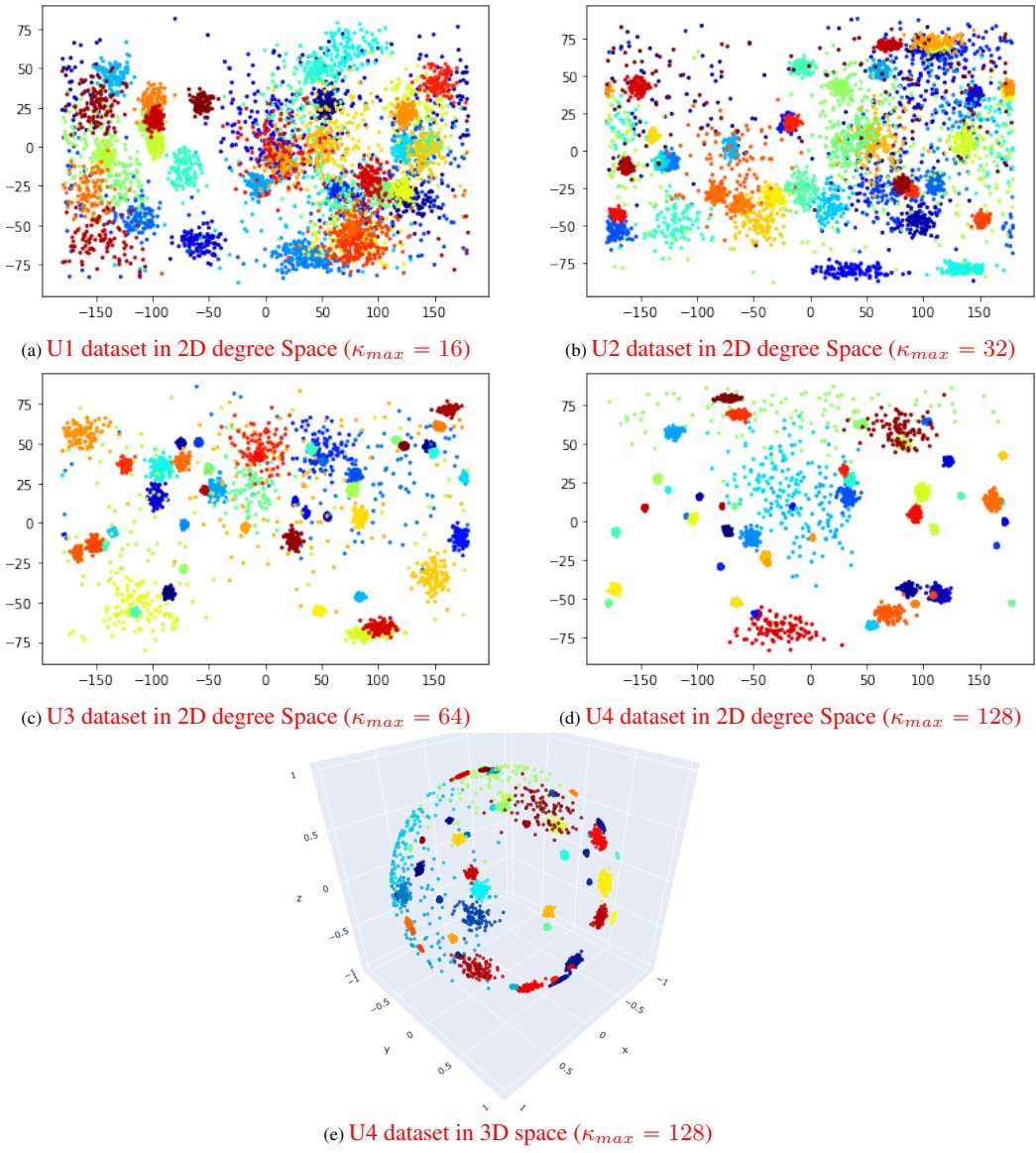

(a) U1 dataset in 2D degree Space ($\kappa_{max} = 16$)

(b) U2 dataset in 2D degree Space ($\kappa_{max} = 32$)

(c) U3 dataset in 2D degree Space ($\kappa_{max} = 64$)

(d) U4 dataset in 2D degree Space ($\kappa_{max} = 128$)

(e) U4 dataset in 3D space ($\kappa_{max} = 128$)

Figure 13: The data distributions of four synthetic datasets (U1, U2, U3, and U4) generated from the uniform sampling method. (e) shows the U4 dataset in a 3D Euclidean space. We can see that if we treat these datasets as 2D data points as $grid$ and $theory$, the $vMF$ distributions in the polar areas will be stretched and look like 2D aniostropic multivariate Gaussian distributions. However, this kind of systematic bias can be avoided if we use a spherical location encoder as *Sphere2Vec*.

Figure 14 visualizes the data distributions of four synthetic datasets with stratified sampling method. They have different $N_\mu$ but the same $\kappa_{max}$. We can see that when $N_\mu$ increases, a more fine-grain stratified sampling is carried out. The resulting dataset has a larger data bias toward the polar areas.

### 9.15.2 SYNTHETIC DATASET EVALUATION RESULTS

We evaluate all baseline models as well as $sphereM+$ on those generated 20 syhthetic datasets as described above. For each model, we do grid search on their hyperparameters for each dataset including supervised learning rate $\eta_{super}$, the number of scales $S$, the minimum scaling factor $r_{min}$, the number of hidden layers and number of neurons used in $\mathbf{NN}_{ffn}(\cdot) - h$ and $k$. The best performance of each model is reported in Table 8. Some observations can be made from Table 8:

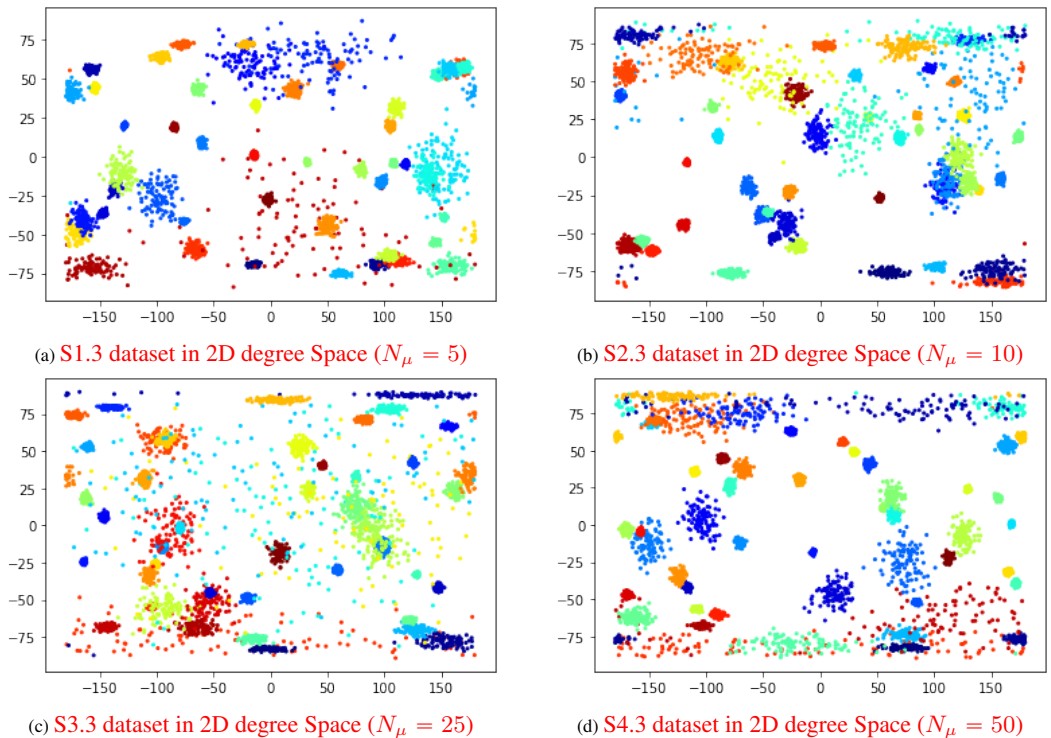

(a) S1.3 dataset in 2D degree Space ($N_\mu = 5$)

(b) S2.3 dataset in 2D degree Space ($N_\mu = 10$)

(c) S3.3 dataset in 2D degree Space ($N_\mu = 25$)

(d) S4.3 dataset in 2D degree Space ($N_\mu = 50$)

Figure 14: The data distributions of four synthetic datasets (S1.3, S2.3, S3.3, and S4.3) generated from the stratified sampling method with $\kappa_{max} = 64$. We can see that when $N_\mu$ increases, a more fine-grain stratified sampling is carried out. The resulting dataset has a larger data bias toward the polar areas.

1. $sphereM+$ is able to outperform all baselines on all 20 synthetic datasets. The absolute Top1 improvement can go up to 2% and the error rate deduction can go up to 30.8%. This shows the robustness of $sphereM+$.

2. When the dataset is fairly easy to classify (i.e., all baseline models can produce 95+% Top1 accuracy), $sphereM+$ is still able to further improve the performance and gives a very large error rate reduction (up to 30.8%). This indicates that $sphereM+$ is very robust and reliable for datasets with different distribution characteristics.

3. Comparing the error rate of different stratified sampling generated datasets (S1.j - S4.j) we can see that when we keep $\kappa_{max}$ fixed and increase $N_\mu$, the relative error reduction $ER$ become larger. Increasing $N_\mu$ means we do a more *fine-grain* stratified sampling. The resulting datasets should sample more $vMF$ distributions in the polar regions. This phenomenon shows that **when the dataset has a larger data bias towards the polar area, we expect $sphereM+$ will be more effective.**

Table 8: Compare $sphereM+$ to baselines on synthetic datasets. U1 - U4 indicate 4 synthetic datasets generated based on the uniform sampling approach (see Appendix 9.15.1). S1.1 - S4.4 indicate 16 synthetic datasets generated based on the stratified sampling apprach. For all datasets have $\mathcal{C} = 50$ and $\mathcal{SP} = 100$. For each model, we perform grid search on its hyperparameters for each dataset and report the best Top1 accuracy. The $\Delta Top1$ column shows the absolute performance improvement of $sphereM+$ over the best baseline model (bolded) for each dataset. The $ER$ column shows the relative reduction of error compared to the best baseline model (bolded). We can see that $sphereM+$ can outperform all other baseline models on all of these 20 synthetic datasets. The absolute Top1 accuracy improvement can be as much as 2.0% for datasets with lower precisions, and the error rate deduction can be as much as 30.8% for datasets with high precisions.

| ID | Method | $N_\mu$ | $C_\mu$ | $\kappa_{min}$ | $\kappa_{max}$ | $xyz$ | $wrap$ | $wrap+ffn$ | $rff$ | $rbf$ | $grid$ | $theory$ | $sphereM+$ | $\Delta Top1$ | $ER$ |
|---|---|---|---|---|---|---|---|---|---|---|---|---|---|---|---|
| U1 | uniform | - | - | 1 | 16 | 67.2 | 67.0 | 66.9 | 66.8 | 46.6 | **68.6** | 67.8 | **69.2** | 0.6 | -1.9 |
| U2 | | | | | 32 | 73.1 | 75.1 | 73.9 | 72.3 | 58.4 | 76.2 | **76.5** | **77.4** | 0.9 | -3.8 |
| U3 | | | | | 64 | 86.1 | 90.1 | 88.3 | 89.0 | 91.7 | 92.3 | **92.7** | **93.3** | 0.6 | -8.2 |
| U4 | | | | | 128 | 91.8 | 94.9 | 92.3 | 92.5 | 97.4 | 97.5 | **97.7** | **98.0** | 0.3 | -13.0 |
| S1.1 | stratified | 5 | 10 | 1 | 16 | 68.7 | 69.7 | 68.8 | 68.6 | **70.5** | 69.5 | 69.4 | **72.3** | 1.8 | -6.1 |
| S1.2 | | | | | 32 | 76.7 | 79.1 | 78.1 | 78.4 | 81.1 | **81.2** | 79.2 | **82.9** | 1.7 | -9.0 |
| S1.3 | | | | | 64 | 91.2 | 92.5 | 92.9 | 92.6 | 94.7 | 94.8 | **94.9** | **95.4** | 0.5 | -9.8 |
| S1.4 | | | | | 128 | 86.5 | 91.6 | 88.3 | 92.4 | 93.5 | **95.2** | 94.9 | **96.1** | 0.9 | -18.7 |
| S2.1 | | 10 | 5 | 1 | 16 | 70.5 | 71.3 | 70.7 | 70.4 | 46.6 | **72.0** | 70.7 | **74.0** | 2.0 | -7.1 |
| S2.2 | | | | | 32 | 76.1 | 79.7 | 78.2 | 78.6 | 61.2 | **80.9** | 80.5 | **82.3** | 1.4 | -7.3 |
| S2.3 | | | | | 64 | 88.0 | 89.9 | 88.2 | 88.5 | 80.0 | **92.5** | 91.9 | **93.3** | 0.8 | -10.7 |
| S2.4 | | | | | 128 | 94.4 | 96.6 | 96.7 | 95.5 | 94.0 | **97.6** | 97.6 | **98.1** | 0.5 | -20.8 |
| S3.1 | | 25 | 2 | 1 | 16 | 66.2 | 66.3 | 64.7 | 65.6 | **67.1** | 66.7 | 66.7 | **68.3** | 1.2 | -3.6 |
| S3.2 | | | | | 32 | 80 | 82.5 | 80.7 | 81.6 | 83.4 | **84.5** | 82.1 | **85.9** | 1.4 | -9.0 |
| S3.3 | | | | | 64 | 85.4 | 86.0 | 85.7 | 86.2 | 89.1 | **89.6** | 88.6 | **91.0** | 1.4 | -13.5 |
| S3.4 | | | | | 128 | 93.2 | 96.0 | 94.8 | 95.7 | 97.2 | 97.3 | **97.4** | **98.0** | 0.6 | -23.1 |
| S4.1 | | 50 | 1 | 1 | 16 | 64.8 | **67.4** | 66.0 | 66.3 | 66.9 | 67.1 | 64.5 | **68.4** | 1 | -3.1 |
| S4.2 | | | | | 32 | 75.6 | 78.2 | 77.4 | 77.4 | 78.4 | **80.1** | 78.3 | **81.0** | 0.9 | -4.5 |
| S4.3 | | | | | 64 | 91.3 | 93.9 | 93.7 | 93.8 | 95.0 | **95.2** | 94.0 | **96.1** | 0.9 | -18.7 |
| S4.4 | | | | | 128 | 94.3 | 95.5 | 94.4 | 94.7 | 95.4 | **97.4** | 96.5 | **98.2** | 0.8 | -30.8 |

