# OpenReview forum: "Sphere2Vec: Self-Supervised Location Representation Learning on Spherical Surfaces"
_ICLR.cc/2022/Conference — ICLR 2022 Submitted_

### Official Review · Reviewer_toM9 · 2021-10-20

**Correctness:** 3
**Technical Novelty And Significance:** 2
**Empirical Novelty And Significance:** 2
**Recommendation:** 5
**Confidence:** 4

**Main Review:**

Strengths:

It's interesting to have a positional encoding method for spherical coordinates, which could have potentials for many other tasks.

Weaknesses:

1) Though the idea is interesting,  I would propose an alternative for embedding GPS coordinates. Would you please discuss it or add it as a baseline?

Given GPS coordinates, we can first transform them to an Earth-centered, Earth-fixed coordinate system, which is a Cartesian coordinate system. Using a simple MLP or (sin/cos) to encode these Cartesian coordinates to the network and check the performance.

2) From Table 1, it seems the proposed method marginally improves previous methods.

3) From the introduction section and Figure 2, it seems the assumption of "assume I and x are conditionally independent given y" does not hold since I and x co-exist.

4) Equation 1. Why do we need to enforce that "multi-scale representation of x preserves the spherical surface distance"? Would you please add some motivations? From the tasks (Figure 2) listed in this paper, I would assume the GPS coordinates are distributed sparsely. Pairwise spherical surface distance may vary on a large scale and is noisy. Do we really need this constraint?

**Summary Of The Paper:**

This paper proposes a positional embedding method for GPS coordinates, which lie on a sphere.

Based on the Double Fourier Sphere, this paper proposes a family of encoding methods and show their effectiveness.

This paper also gives a family of loss functions to enable self-supervised training.

**Summary Of The Review:**

While I like this work and it proposes an interesting positional encoding method for spherical coordinates, I would like to see the authors' reply for the weaknesses to make further decisions.

---

> ### Author Response · Authors · 2021-11-23
> **Main Response to Reviewer toM9**
>
> We would like to thank Reviewer toM9 for the thoughtful comments. Please refer to our main response and our updated submission.
>
> 1. **Add another baseline which first convert (lat, lon) to a 3D Cartesian coordinate system before feeding into MLP**
>
> Thanks for the interesting baseline. We have added this model as one of our baseline. It performs similarly to wrap* (see the main response). If we further apply sin/cos to the coordinates it will be equivalent to one of our proposed variations, sphereC in Appendix 9.2, without multi-scale representation.
>
> 2. **Proposed models only marginally improve previous models.**
>
> A. We add Table 4 about the model performance’s standard deviations. We show that Sphere2Vec can outperform all baselines on all datasets. The performance differences between sphere2vec and the baselines (e.g., grid) are statistically significant. (see the main response).
>
> B. We have added experiments with synthetic datasets for an in depth analysis of the impact of spherical coordinates under controlled settings (see the main response).
>
> C. The performance improvement is comparable to those from previous works (See the main response and Appendix 9.9).
>
> D. Even though our approach is called sphere2vec, we believe that a major contribution of this work is the self-supervised training loss, which is critical for data limited settings as demonstrated in Figure 5.
>
>
> 3. **"assume I and x are conditionally independent given y"**
>
> This independent assumption has been adopted by multiple previous works such as Chu et al, 2019, Mac Aodha et al, 2019, Mai et al, 2020b. This assumption is suboptimal, since the fusion of image and location information is at the very end of the pipeline. However, it makes the development and experiment much easier, since the image and location models can be trained independently. Early fusion of image and location information is a promising direction, which we'd like to explore in the near future.
>
>
> 4. **Spherical distance preservation constraints**
>
> Please see more discussion in our main response for the motivation of distance preservation, and further comparison between grid and sphere2vec. The difference between grid and sphere2vec in the synthetic dataset experiments (Table 8) helps to demonstrate the impact of this property.

---

> ### Author Response · Authors · 2021-12-03
> **Discussion**
>
> Dear Reviewer toM9, please let us know if our response addressed your concerns or if you have any further questions.

---

### Official Review · Reviewer_vjnw · 2021-10-30

**Correctness:** 4
**Technical Novelty And Significance:** 3
**Empirical Novelty And Significance:** 3
**Recommendation:** 6
**Confidence:** 4

**Main Review:**

Positives

  - how to integrate contextual information, e.g., location, as part
    of vision pipelines is an interesting and challenging problem, with
    demonstrated performance improvements on various tasks (e.g., fine-grained
    species classification)

  - the authors propose a novel multi-scale method for encoding
    spherical coordinates (several variants) based on Double Fourier
    Sphere, including an analysis of the distance preserving quality in
    the supplemental

  - results show consistent improvements over baselines on seven different
    datasets

  - extensive experiments, including ablation studies of various
    components in the supplemental material

  - the self-supervised component, for pretraining the location
    encoder using unlabeled data, is shown to be useful in scenarios
    where there is limited labeled data


Negatives

  -  grammatical errors throughout

    For example, I believe the first contribution in Section 1 should be "distance preserving" instead of "distant reserving".

  - missing details

    Section 3.2 has some elements whose description could be expanded
    on. For example, the notation for omitting terms of the loss
    function. Another example would be the linear projection layer
    that is mentioned only once in passing (and shown in Figure 1).
    Further, implementation details (optimizer, learning rate schedule,
    hardware, etc.) seem to be missing or relegated to the supplemental.

  - though the results are impressive, it is not clear from the
    results which version of Sphere2Vec (from the proposed variants)
    should be used in which scenario (in other words, there is no
    consistent winner)

  - the self-supervised component of the proposed approach
    appears to be limited to a pretraining strategy, with results
    showing it useful only when there is limited labeled data

**Summary Of The Paper:**

This paper explores developing location encodings for spherical
surfaces and their application to tasks in geo-aware image
classification. The authors propose a method for encoding spherical
locations into a higher dimensional space (based on Double Fourier
Sphere) that preserves distances. The proposed method, referred to as
Sphere2Vec, is demonstrated for three tasks and evaluated using seven
datasets. In addition, a self-supervised method is proposed for using
unlabeled data to pretrain the location encoder.  This self-supervised
pretraining stage is shown to improve performance, especially when
available labeled data is limited.

**Summary Of The Review:**

  This paper explores two important problems: 1) given a geotagged
  image, how to encode location information and incorporate it as
  context in existing image processing pipelines, and 2) how to take
  advantage of a large corpus of images with known geolocation, that
  are not annotated (unlabeled) for the primary task. Specifically,
  this paper explores how to encode geospatial information in the form
  of spherical coordinates.

  For several tasks in geo-aware image classification, the
  proposed multi-scale spherical location encoder outperforms
  baselines. However, the proposed self-supervised method for
  incorporating unlabeled geotagged imagery is limited to a pretraining
  strategy when there is limited labeled data. Additionally, several
  variants of the method are proposed, and it is not clear what variant
  to use when.

  Ultimately, I think this paper tackles an important problem and
  demonstrates the general applicability of the method. While the
  manuscript has minor errors throughout and some missing details, I
  lean towards accept for my initial rating.

---

> ### Author Response · Authors · 2021-11-23
> **Main Response to Reviewer vjnw**
>
> We would like to thank Reviewer vjnw for the thoughtful comments. Please refer to our main response and our updated submission.
>
> 1. **Which version of Sphere2Vec should be used in which scenario (no consistent winner)**
>
> A. Amongst the 7 datasets we have worked with, sphereM+ performs the best on 4 datasets while showing competitive performance on the other 3 (the 2nd best model). So we suggest using sphereM+ as the top 1 candidate for spherical location encoding.
>
> B. SphereM+ features are a superset of that of sphereC, sphereC+, and sphereM, but SphereM+ is still more parameter efficient than sphereDFS.
>
> 2. **Self-supervised component of the proposed approach appears to be limited to a pretraining strategy**
>
> Self-supervised training and few-shot learning EW very important in practice, especially for geospatial tasks, since collecting labels for geographical images (such as remote sensing images) is both expensive and time consuming. For example, the widely-used SpaceNet 7 dataset only has labels for 60 locations around the globe, and a good few-shot learning capability becomes very critical to the task.
>
>
> 3. **Missing details in Section 3.2**
>
> We added model implementation details about MLP layers, optimizer, learning rate schedule, hardware, and etc in Appendix 9.7.
> We moved important details from the appendix to the main text (e.g., the definition of "simcse" loss).
>
>
> 4. **Grammatical errors**
>
> Sorry for the presentation. We have improved the wordings, and will continue improving.

---

> > ### Comment · Reviewer_vjnw · 2021-11-29
> > **thoughts**
> >
> > Thanks for the response.
> >
> > I have reviewed the updated paper as well as the other reviewers comments and author responses. I think the authors have done a good job addressing the majority of the concerns and I continue to be positive.

---

### Official Review · Reviewer_6cCW · 2021-11-01

**Correctness:** 3
**Technical Novelty And Significance:** 2
**Empirical Novelty And Significance:** 2
**Recommendation:** 3
**Confidence:** 4

**Main Review:**

Positives:

- I think the problem setting, geo-aware image classification, is interesting and I haven't seen a lot of work exploring the issue of encoding geographic location, which has some clear challenges.
- There are quite a few datasets and baselines used in the comparison.

Negatives:

- Table 1 shows that the effect size is relatively small. The proposed approach isn't clearly worse than others, but it doesn't seem to make a major difference.

- The core method is composed of several components (the initial encoding, the MLP, and self-supervised pre-training). However, the only thing that is varied is the initial encoding. This leaves open the question of how changes to the other components would impact the metrics. It could be that a slightly deeper or shallower MLP would change which method is the best. The paper claims that many aspects were fixed to provide a fair evaluation, but this often hides the fact that the proposed contribution is only superior given the particular settings, and not that significant overall.

- It is claimed that "grid ... performs poorly at a global scale due to its inability to preserve spherical distances" and provides a proof. However, the quantitative metrics show that grid performs about as well as the proposed method.

- The presentation needs significant improvement. Some figures are too small (Fig 4) or have labels that are too small (Fig 5). Also, see below for grammar issues. There are important details that aren't described well, for example, what exactly is the MLP? How sensitive are the results to the hyperparameters of the MLP?

- Section 5.1: it is argued that "wrap and grid have over generalization issue in data sparse area" and that the proposed method provides better results. It's unclear why this is the case. Much of the theoretical justification relies on the fact that the proposed approach preserves spherical distances, but the connection between that preservation and the differences between the methods isn't made strongly enough.

Minor Comments:

- I would have liked to have seen a comparison to the conversion of lat/lon into x,y,z coordinates using something like earth-centered, earth fixed coordinates.

- There are numerous small grammatical errors. They didn't distract from clarity, but it needs significant improvement. Here are a few from the paper (but there are others and quite a few in the supplemental as well):
   - "may provide clue to their real classes" -> "may provide a clue to their real classes"
   - "data parse areas" -> "data sparse areas"
   - "Despite of its effectiveness" -> "Despite its effectiveness"
   - "distance reserving" -> "distance preserving"
   - "identities base on their" -> "identities based on their"
   - "are public available" -> "are publicly available"
   - There are many more in the supplemental.

- It seems like there are too many important details left to the appendix, including defining a key loss component "simcse", which isn't defined until 9.11.

- It might be worth including a reference to http://bmvc2018.org/contents/papers/0586.pdf (Section 3.1), which proposes an approach to learning a location embedding in conjunction with geo-tagged ground-level images. This is like the first stage of training in the proposed method but requires a known timestamp.


**Summary Of The Paper:**

This paper proposes a strategy for learning a location embedding (i.e., translating lat/lon into a useful high-dimensional vector). The idea is to first encode the location into the double Fourier sphere basis (or a subset) and then pass this through a multilayer perceptron. Several alternatives to the first stage are evaluated.

Two stages of training are proposed:
1) Weakly Supervised Pre-Training: use a contrastive loss to train the embedding by comparing it to features extracted from geotagged images. The idea is that a good location embedding should be sufficient to recognize whether a given image was captured in that location.
2) Supervised Fine-Tuning: given a geotagged image, extract image and location features, combine them to predict the image classification label.

The approach is demonstrated on several geo-aware image classification applications. The idea with these applications is that combining image features with the location embedding is sometimes enough to resolve ambiguity from the image alone.

**Summary Of The Review:**

I don't think this paper is suitable in its current state. The experimental results show a small effect size and it's unclear how stable the results are to changes in the MLP, and other components. There is also a weak connection between some of the stated benefits (preserving spherical distances) and the actual benefits. It's entirely possible that the proposed methods are better in some settings for some other reason.

---

> ### Author Response · Authors · 2021-11-23
> **Main Response to Reviewer 6cCW**
>
> We would like to thank Reviewer 6cCW for the thoughtful comments. Please refer to our main response and our updated submission.
>
> 1. **reference and comparison to image localization (Zhai et al. 2018)  https://arxiv.org/abs/1909.07499**
>
> We thank the suggestion of this relevant work and have updated our related work. Their setting is very similar to us -- learning location representation from unlabeled image-location pairs. However, one main difference and also one of our main contributions of space2vec is its contrastive self-supervised loss. Zhai et al. (2018) applied cross entropy loss for discretized  location (or time), which cannot leverage the continuity of the function to be approximated.
>
>
> 2. **The effect size is relatively small. … grid performs about as well as the proposed method**
>
> A. We add Table 4 about the model performance’s standard deviations. We show that Sphere2Vec can outperform all baselines on all datasets. The performance differences between sphere2vec and the baselines (e.g., grid) are statistically significant. (see the main response).
>
> B. We have added experiments with synthetic datasets for an in-depth analysis of the impact of spherical coordinates under controlled settings (see the main response).
>
> C. The performance improvement is comparable to those from previous works (See the main response and Appendix 9.9).
>
>
> 3. **What exactly is the MLP? How sensitive are the results to the hyperparameters of the MLP?**
>
> A. We add the implementation details about MLP in Appendix 9.7 and the model training details in Appendix 9.8.
>
> B. We experimented with different numbers of MLP layers h, and different sizes of hidden layers k. We can see (from the updated Table 4) that the performance is not sensitive to k, while the best h is usually just 1.  (see the main response)
>
>
> 4. **There is also a weak connection between some of the stated benefits (preserving spherical distances) and the actual benefits.**
>
> A. For the motivation of distance preservation please see more discussion in our main response.
>
> B. The difference between grid and sphere2vec  in the synthetic dataset experiments  (Table 8) helps to demonstrate the impact of this property.
>
>
> 5. **The connection between that preservation and the differences between the methods isn't made strongly enough. Why do "wrap and grid have over generalization issues in data sparse areas"?**
>
> A. One issue with wrap, wrap+ffn is the lack of multi-scale representation. In other words, they directly pass coordinates into the network (MLP) without preprocessing with a Fourier input mapping as grid and Sphere2Vec do. Many previous studies (https://arxiv.org/abs/2006.10739, https://arxiv.org/abs/2003.04560, https://arxiv.org/abs/1806.08734) have shown that MLPs have difficulty learning high-frequency functions, a phenomenon referred to as “spectral bias”, whereas preprocessing the input with a Fourier feature mapping (such as grid and Sphere2Vec) enables the MLP to represent higher frequency details. The reason why wrap produces overgeneralized results is because of the so-called spectral bias.
>
> B. As for grid, although it utilizes a Fourier input mapping, this mapping is designed for the 2D Euclidean space. So compared with Sphere2Vec, grid will perform poorly in the polar areas because there is a large distortion in the polar area. Our experimental results in Figure 4c and 4d also verify our argument.
>
> C. We added an experiment on synthetic data (Appendix 9.15) to further demonstrate the difference between grid and sphere2vec under controlled settings (see the main response for details).
>
>
> 6. **A comparison to the conversion of lat/lon into x,y,z coordinates**
>
> Thanks for the interesting baseline. We have added the result of xyz representation in table 1. It performs similarly to wrap* since both of them lack multi-scale encoding (see the main response).
>
>
> 7. **Presentations**
>
> We have moved important details from the appendix to the main text (e.g., the definition of "simcse" loss).
>
> We are sorry that Figure 4 is rather small. This is due to the page limit. If we can get our paper accepted and have one extra page, we will enlarge figure 4 and also add synthetic data results into the main paper.

---

> > ### Comment · Reviewer_6cCW · 2021-11-29
> > **thoughts**
> >
> > I thank the authors for the extensive responses and updates to the manuscripts. This clearly represents a significant amount of work. I am, however, still concerned about a fundamental issue that I raised in my initial review. At the core, the issue is the limited effect size and lingering doubts about the hyperparameter search. As was explained by the authors in this comment (https://openreview.net/forum?id=FS0XKbpkdOu&noteId=M1rrTWceb0k), there was quite a bit more tuning done for the proposed approaches than for some of the baselines. In addition, it doesn't look like any search over a weight decay hyperparameter was performed. This is especially concerning given that it sounds like the models were very quick to reach a point where additional training was rendered useless due to overfitting. I still worry that what we are seeing is that the proposed approach is marginally better given the peculiarities of the hyperparameter search, but might not be better with a moderately improved hyperparameter tuning of some of the baselines.

---

> > > ### Author Response · Authors · 2021-11-30
> > > **Experiments on weight decay**
> > >
> > > We thank Reviewer 6cCW for his/her thoughtful comments. We are sorry that we have not made it clearer.
> > >
> > > 1. **it doesn't look like any search over a weight decay hyperparameter was performed. ... it sounds like the models were very quick to reach a point where additional training was rendered useless due to overfitting**
> > >
> > > Thanks a lot for the suggestion of weight decay. In fact, we did not use weight decay in our experiment with Adam optimizer, since none of the previous work or our baselines use weight decay.
> > > Here we add more experiments to test the effectiveness of weight decay. We vary the weight decay, when training location encoders. Because of the time limitation, we only test the result of theory, grid, and sphereM+ on the BirdSnap† dataset. We trained each model for at least 100 epoches since it takes longer to converge with weight decays. The results are shown below.
> > >
> > > | weight decay | BirdSnap† |       |          |
> > > |--------------|-----------|-------|----------|
> > > |              | theory    | grid  | sphereM+ |
> > > | 0            | 79.24     | 79.44 | 80.34    |
> > > | 0.00001      | 79.62     | 79.75 | 79.71    |
> > > | 0.00005      | 79.8      | 79.66 | 80.38    |
> > > | 0.0001       | 79.71     | 79.8  | 80.68    |
> > > | 0.001        | 79.35     | 79.31 | 78.87    |
> > > | 0.01         | 77.06     | 77.1  | 76.17    |
> > >
> > > We can see that an appropriate weight decay does further improve the performance (Top1 accuracy) of all these three models by ~0.3%. However, the conclusion is still the same -- SphereM+ can outperform those two strongest baselines, since the comparison among different approaches is still a fair comparison.
> > >
> > >
> > >  2. **There was quite a bit more tuning done for the proposed approaches than for some of the baselines.**
> > >
> > > In fact, all baselines as well as all our proposed Sphere2Vec models follow exactly the same hyperparameter tuning strategy (See https://openreview.net/forum?id=FS0XKbpkdOu&noteId=M1rrTWceb0k). We did not do more tuning on our proposed approaches. Since different models have different numbers of hyperparameters, their tuning processes may look different. However, grid and theory follow the exact same tuning process as Sphere2Vec models. We can still show that our model can outperform grid and theory.
> > >
> > > Based on our experiments, the strongest baselines are grid and theory model (Mai 2020b). This can be seen from Table 1 (7 real-world datasets) and Table 8 (20 synthetic datasets). This makes sense because compared with other baselines, grid and theory add a multi-scale Fourier input mapping layer. And multiple recent studies (https://arxiv.org/abs/2006.10739, https://arxiv.org/abs/2003.04560, https://arxiv.org/abs/1806.08734, https://arxiv.org/abs/1909.05215) have shown that adding a Fourier input mapping layer can improve the model performance.
> > >
> > > As for wrap, wrap+fnn, xyz, rbf, and rff, since these baseline models have fewer hyperparameters (they do not have Fourier input mapping layer), it looks like they are tuned less. However, in reality, we follow the same hyperparameter tuning strategy.
> > >
> > > Moreover, in Table 1, we compared our results with the reported resulted from the previously published work such as wrap* (Mac Aodha et al., 2019) on all first 6 datasets, grid and theory’s result on BirdSnap†, NABirds† (Mai et al. 2020b), and  Ayush et al. (2020) results on fMoW. In fact, we have tuned the wrap model with our protocol, and the results we obtained (wrap in Table 1) are even better than the results reported by Mac Aodha et al. (2019) (wrap* in Table 1). This also shows that our hyperparameter tuning on the baseline models is quite extensive.
> > >
> > >
> > > 3. **the effective size of the task**
> > >
> > > The effective size is determined by the tasks in our study, and our effect size is comparable to the results from previous work (Mac Aodha et al, 2019, Mai et al, 2020b). Table 4 shows the standard deviations of each model, from which we can compute that the improvements from Spere2Vec over baseline models are statistical significance. As mentioned earlier (https://openreview.net/forum?id=FS0XKbpkdOu&noteId=M1rrTWceb0k), the histogram of all each model with different hyperparameter settings (https://imgur.com/a/jLnSLXt) also shows that the performance of sphereM+ is clearly distinguishable from other models.

---

> > > ### Author Response · Authors · 2021-12-03
> > > **Discussion**
> > >
> > > Dear Reviewer 6cCW, please let us know if our response addressed your concerns or if you have any further questions.

---

### Official Review · Reviewer_6Te4 · 2021-11-03

**Correctness:** 3
**Technical Novelty And Significance:** 1
**Empirical Novelty And Significance:** 2
**Recommendation:** 5
**Confidence:** 2

**Main Review:**

Strengths:
1) The approach is demonstrated to be better than the previous method both qualitatively and quantitatively.
2) Description of the losses and algorithms were fairly complete with minor errors.
3) Good references on topics beyond CV/DL papers.


Weaknesses:
1) Paper was slightly harder read and some useful information such as the definition of the studied approaches were left in the appendices. Reordering the information in the text would help here.
2) Most of the gains come from the framework over the spherical coordinate approach. This seems reasonable as using spherical coordinates over euclidean coordinates doesn't seem to be something a few layers of networks can't learn easily. 3
3) Some errors in the text:
    a) In practice it help with --> helps with
    b) Eq. 6: second term in the second line seems to have errors
    c) Supervised Fine-Turning --> Fine-tuning
    d) \delta MRR was not defined in the text



**Summary Of The Paper:**

The paper studies how to use location information (geographical location) of images to help with image classification tasks. The paper studies use of spherical based location over GPS lat long and argues that the euclidean space does not preserve distances and has distortions whereas a spherical coordinate approach would not have the inherent problem. The approach demonstrates a framework to train the location encoding in an unsupervised way to with losses based on representation learning approaches. The paper studies multiple ways of expanding the spherical coordinates and evaluates their performance and empirically demonstrate the strength in their performance.

**Summary Of The Review:**

Paper proposes a framework and uses spherical coordinates over encoding method to help with geo-aware image classification. The paper propose the title of sphere2vec for novelty but the main improvements to the method is not with the encoding but with the framework itself. For readers, it would be a slightly disappointing results if the reader is reviewing spherical coordinate as an approach. As a paper, the results does show improvements through the framework, in the form of representation learning. Thus, the recommendation is a marginally below acceptance threshold.

---

> ### Author Response · Authors · 2021-11-23
> **Response to Reviewer 6Te4**
>
> We would like to thank Reviewer 6Te4 for the thoughtful comments. Please refer to our main response and our updated submission.
>
> 1. **A slightly disappointing results if the reader is reviewing spherical coordinate as an approach**
>
> A. We add Table 4 about the model performance’s standard deviations. We show that Sphere2Vec can outperform all baselines on all datasets. The performance differences between sphere2vec and the baselines (e.g., grid) are statistically significant. (see the main response).
>
> B. We have added experiments with synthetic datasets for an in-depth analysis of the impact of spherical coordinates (see the main response).
>
> C. The performance improvement is comparable to those from previous works (See the main response and Appendix 9.9).
>
> D. Even though our approach is called sphere2vec, we believe that a major contribution of this work is the self-supervised training loss, which is critical for data-limited settings as demonstrated in Figure 5.
>
> 2. **Some useful information such as the definition of the studied approaches were left in the appendices**
>
> We have updated the paper to have definitions in the main text.
>
>
> 3. **Some errors in the text**
>
> We have updated the paper to fix the errors.

---

> ### Author Response · Authors · 2021-12-03
> **Discussion**
>
> Dear Reviewer 6Te4, please let us know if our response addressed your concerns or if you have any further questions.

---

### Author Response · Authors · 2021-11-23
**Main reponse to All Reviewers**

We thank the reviewers for their insightful comments. We have updated our submission by following each reviewer's suggestions. We have added Appendix 9.7, 9.8, 9.9, and 9.15. All changes are highlighted in red.

Here we address the common issues among the reviews.

1. **Result with Synthetic Data**

We have added experiments with synthetic datasets for an in-depth analysis of the impact of spherical coordinates (see Appendix 9.15 and Table 8). We show that sphereM+  can outperform all other baseline models on all 20 synthetic datasets. The absolute Top1 accuracy improvement can be as much as 2% for datasets with low prediction precisions, and the relative error rate deduction can be as much as 30% for datasets with high prediction precisions. Please refer to Appendix 9.15 for a detailed discussion.


2. **The importance of self-supervised training**

Even though our approach is called Sphere2Vec, we believe that a major contribution of this work is the self-supervised training loss, which is critical for data-limited settings (i.e., few-shot learning) as demonstrated in Figure 5 and Appendix 9.11. Self-supervised pre-training and few-shot learning are very important in practice, especially for geospatial tasks (Ayush et al., 2021), since collecting labels for geographical images (such as remote sensing images) is both expensive and time-consuming. For example, the widely-used SpaceNet 7 dataset only has labels for 60 locations around the globe, and a good few-shot learning capability becomes very critical to the task.


3. **Statistical Significance**

Several reviewers pointed out that the performance improvement of Sphere2Vec is rather small and they are not sure whether this improvement is statistically significant. To answer this question, we computed standard deviations (Table 4 page 20) of each model based on multiple runs (5 times) with the reported best hyperparameters. The performance differences between sphere2vec and the baselines (e.g., grid) are statistically significant.

Moreover, we need to point out that these performance improvements are comparable to those from the previous studies on the same tasks. This is due to the nature of the used datasets. For example, Mai et al. (2020b) showed that grid/theory has 0.79%, 0.44% absolute Top1 accuracy improvement on BirdSnap† and NABirds† dataset respectively. Mac Aodhaet al. (2019)  showed that wrap has 0.09%, 0%, 0.04% absolute Top1 accuracy improvement on BirdSnap, BirdSnap†, and NABirds† dataset.


4. **MLP ablation study**

We are sorry that MLP ablation study details were not clearly described in the previous version. We experimented with different numbers of MLP layers h and different sizes of hidden layers k.  The best hyperparameter combinations for Sphere2Vec are shown in Table 2. We also add Table 4 to show the performance of the same model with different hidden layers. We can see from Table 4 (page 20) that a deeper MLP does not necessarily lead to better performance. In fact, the performance of location encoders is not sensitive to h, while the best h is usually just 1. In other words, the systematic bias (i.e., distance distortion) introduced by grid and theory can not later be compensated by a deep MLP.


5. **3D coordinate baseline**

We have added this model as one of our baselines (xyz in the updated Table 1). In fact, xyz can be seen as a special case of sphereC model when the number of scales S = 1.  Sphere2Vec models can outperform xyz with a significant margin.  xyz performs similarly to wrap* since both of them lack multi-scale encoding.


6. **The motivation of distance preservation**

We introduced the distance preservation constraint in order to promote the smoothness of the transformed function. In machine learning, this smoothness is also the idea behind many kernel-based methods. In geoscience, smoothness corresponds to the First Law of Geography - "everything is related to everything else, but near things are more related than distant things."

In our cases, the species occurrences have various densities. For example, plant species are clustered together, while large animals such as tigers and bears are usually living by themselves. It is true that the distribution of a species can be non-continuous at large scales, but we can always find a small enough scale, in which the species distribution is locally continuous. This is also the motivation behind multi-scale representation learning.

Our synthetic dataset experiments show that if we treat these spherical data points as 2D data points (as in grid and theory), vMF distributions in the polar areas can be stretched to very extended shapes (see Figure 13 d, Figure 14 c,d) making model learning more difficult. However, this kind of systematic bias can be avoided if we use a spherical-distance-preserving location encoder as Sphere2Vec. Moreover, when the dataset has a larger data bias towards the polar area, Sphere2Vec will be more effective.

---

> ### Comment · Reviewer_6cCW · 2021-11-24
> **understanding the hyperparameter search**
>
> I continue to be skeptical about the significance of these results. Given the small differences in performance, little details in how the models were trained become important to me.
>
> I've got a few clarifying questions:
>
> If I understand correctly from Section 9.8.1, a grid search over hyperparameters was performed. Given the listed hyperparameters, this looks like it would result in 6 (N_super) x 3 (S) x 7 (and so on) x 11 x 3 x 7 x 3 = 87,318 hyperparameter settings. It sounds like this was done independently for each model type and dataset, which adds at least a factor of 10 by my count. Was this actually done? If not, what was done?
>
> How long does it take to train each model? The only thing that I can find that relates to how long the model was trained is as follows "After the model is convergence, we use a new learning rate". What was the convergence criteria?

---

> > ### Author Response · Authors · 2021-11-25
> > **More details about the hyperparameter tuning process and some analysis (Part I)**
> >
> > We thank Reviewer 6cCW for his/her thoughtful comments. We give more details about the hyperparameter tuning process and provide some analysis on that.
> >
> > 1. **Hyperparameter tuning procedure**
> >
> > We are sorry that the hyperparameter tuning part was not clearly explained. Indeed, it is impossible to search all combinations, which is a very large search space. We employ a coordinate ascent strategy -- searching one parameter (or a few parameters) at a time, starting from the most important parameter, and rotate among all parameters until convergence. The hyperparameter tuning was done for each model and each dataset independently. In practice, we tune the hyperparameters for all models on one dataset, and then we go to the next one. Since each run will take small memory and computation, we run all models on the same dataset in parallel to save time.  For most model/datasets, we only rotate the parameter tuning for two rounds to see the convergence.
> >
> > From our past experiment, ReLU and dropout rate 0.5 work well for most of the models and datasets, so we fix them for all our experiments.  In fact, for all models, Sigmoid yields significantly lower results, and the models with LeakyReLU have lower performance than their ReLU counterparts.
> >
> > 1.1  **Gird, theory, and Sphere2Vec models**
> >
> > **For YFCC, BirdSnap, BirdSnap†, and NABirds† datasets**, we do the following hyperparameter tuning steps:
> >
> > A) we first do grid search on those hyparameters which significantly influence the model performance including supervised learning rate \mu_{super} and minimum scaling factor r_{min}, while we keep other hyperparamters fixed (we use initial guesses of 512 as the hidden neurons k, 1 hidden layer, 32 as the number of scales S). In this step, we need to search for 6*7 = 42 models. We pick the top 3 models and move on to the next steps.
> >
> > B) Next, we tune the number of scales, which takes 3 * 3 = 9 models. We pick the top 3 models and go to the next step.
> >
> > C) Finally, for the hyperparameter combinations of these 3 models,  we do grid search by replacing the number of hidden layers h and the number of neurons k of these models. This step takes 3 * 4 * 3 = 36 models. After all these steps, we select the models with the highest performance.
> >
> > **For iNat2017, iNat2018, and fMoW datasets**, it takes a rather long time to train the models. So instead of following the above procedure, we tune hyperparameters one by one in 5 steps by using a beam search like strategy. After each step, the top k choices (k=3) will be used for the tuning of the rest of the hyperparameters in the following steps. The top k choices from the previous step are combined with all possible values of the current hyperparameter to generate candidates. The order in which the parameters are tuned is learning rate \mu_{super}, scaling factor r_{min}, number of scales S, hidden layers h, and number of neurons k.
> >
> >
> > 1.2 **wrap, wrap+ffn, and xyz**
> >
> > As for wrap, wrap+ffn, and xyz, there are no multi-scale encoding parameters. Therefore, we tune the supervised learning rate \mu_{super} first. And then we tune the number of hidden layers h, and the number of neurons k. We amount to 6 +  3*4 = 18 models.
> >
> > 1.3 **rbf, rff**
> >
> > As for rbf, rff, we first tune supervised learning rate \mu_{super}, the kernel size, and the number of kernels  (with initial h = 1, k = 512). Then we pick the top 3 models and tune h and k.
> >
> > 2. **Model performance distributions over the hyperparameter tuning process**
> >
> > To clearly visualize the performance difference between Sphere2Vec and the baseline models, we plot their distributions of Top1 accuracy under different hyperparameter combinations.
> >
> > More specifically, after the hyperparameter tuning process described in the previous section, for each location encoder we got a collection of trained models with different hyperparameter combinations. They correspond to a distribution of Top1 accuracies for this model on the dataset. These figures (https://imgur.com/a/jLnSLXt), compare the distributions of SphereM+ and all baseline models on both BirdSnap† and iNat2018. We can see that the distribution of SphereM+ accuracy is clearly above those of all other baseline models. This further demonstrates the superiority of Sphere2Vec over the baseline models.
> >
> > Please use this link (https://imgur.com/a/jLnSLXt) to see these two figures.

---

> > ### Author Response · Authors · 2021-11-25
> > **More details about the hyperparameter tuning process and some analysis (Part 2)**
> >
> > 3. **Model run time**
> >
> > In terms of the time cost for each model, we provide the average wall time cost of each model on each dataset below. The number indicates the time it takes to run one model until convergence in seconds.
> >
> > Based on the above hyperparameter tuning procedure, we need to run 87 models for the grid, theory, and all Sphere2Vec models on YFCC, BirdSnap, BirdSnap†, and NABirds† dataset. All models need roughly 400 seconds to be trained once. So it roughly takes 10 hours for each model on each dataset. Since we run each model in parallel. So it took us roughly one and half months to get the final results shown in Table 1.
> >
> > | P(y\|x)  | BirdSnap | BirdSnap† | NABirds† | iNat2017 | iNat2018 | YFCC  | fMoW   |
> > |----------------------|----------|-----------|----------|----------|----------|-------|--------|
> > | xyz                  | 63       | 119       | 258      | 1,697    | 1,061    | 270   | 2,586  |
> > | wrap                 | 390      | 815       | 585      | 6,631    | 5,024    | 1,477 | 18,778 |
> > | wrap+ffn             | 267      | 589       | 556      | 5,621    | 4,128    | 744   | 18,811 |
> > | rbf                  | 76       | 291       | 278      | 3,310    | 2,183    | 331   | 19,135 |
> > | rff                  | 118      | 236       | 251      | 2,795    | 1,757    | 542   | 18,876 |
> > | gridcell             | 65       | 200       | 252      | 4,063    | 4,616    | 302   | 18,953 |
> > | theory               | 101      | 242       | 243      | 3,023    | 4,243    | 481   | 19,024 |
> > | sphereC              | 164      | 376       | 335      | 4,374    | 2,573    | 666   | 11,356 |
> > | sphereC+             | 62       | 471       | 369      | 4,343    | 2,656    | 878   | 11,201 |
> > | sphereM              | 257      | 541       | 399      | 4,768    | 3,034    | 1,023 | 11,188 |
> > | sphereM+             | 65       | 164       | 404      | 1,933    | 1,237    | 714   | 10,571 |
> > | sphereDFS            | 388      | 851       | 599      | 5,404    | 3,390    | 1,524 | 20,414 |
> >
> >
> > 4. **Model convergence criteria**
> >
> > In terms of the model coverage criteria, we run each model at most T epochs. We evaluate the model’s performance on the validation and test dataset every 5 epochs. We can observe that the model’s performance on the validation/test dataset will first increase and then drop. To report test performance, we pick the checkpoint corresponding to the highest performance on the validation set.
> >
> > For all dataset except fMoW, the highest epoch will always appear before the 30th epoch. So we train all models for 30 epochs. As for the fMoW dataset, we train each model for 300 epochs to get the highest evaluation score.

---

### Decision · Program_Chairs · 2022-01-20

**Decision:**

Reject

**Comment:**

In spite of some slightly mixed scores (with one borderline positive review), scores are ultimately lukewarm and tend toward negative (and furthermore, reviews are broadly in agreement as to the issues they raise). Main issues center around low significance of the results, and issues with the presentation that need to be addressed.